# Identification of Homeobox Transcription Factors in a Dimorphic Fungus *Talaromyces marneffei* and Protein-Protein Interaction Prediction of RfeB

**DOI:** 10.3390/jof10100687

**Published:** 2024-09-30

**Authors:** Monsicha Pongpom, Nopawit Khamto, Panwarit Sukantamala, Thitisuda Kalawil, Tanaporn Wangsanut

**Affiliations:** 1Department of Microbiology, Faculty of Medicine, Chiang Mai University, Chiang Mai 50200, Thailand; monsicha.p@cmu.ac.th (M.P.);; 2Department of Chemistry, Faculty of Science, Chiang Mai University, Chiang Mai 50200, Thailand; 3Faculty of Dentistry, Chiang Mai University, Chiang Mai 50200, Thailand

**Keywords:** Hox protein, homeobox, homeodomain, molecular dynamics, *Talaromyces marneffei*, morphogenesis, phase transition

## Abstract

*Talaromyces marneffei* is a thermally dimorphic fungus that can cause life-threatening systemic mycoses, particularly in immunocompromised individuals. Fungal homeobox transcription factors control various developmental processes, including the regulation of sexual reproduction, morphology, metabolism, and virulence. However, the function of homeobox proteins in *T. marneffei* has not been fully explored. Here, we searched the *T. marneffei* genome for the total homeobox transcription factors and predicted their biological relevance by performing gene expression analysis in different cell types, including conidia, mycelia, yeasts, and during phase transition. RfeB is selected for further computational analysis since (i) its transcripts were differentially expressed in different phases of *T. marneffei*, and (ii) this protein contains the highly conserved protein-protein interaction region (IR), which could be important for pathobiology and have therapeutic application. To assess the structure-function of the IR region, in silico alanine substitutions were performed at three-conserved IR residues (Asp276, Glu279, and Gln282) of RfeB, generating a triple RfeB mutated protein. Using 3D modeling and molecular dynamics simulations, we compared the protein complex formation of wild-type and mutated RfeB proteins with the putative partner candidate TmSwi5. Our results demonstrated that the mutated RfeB protein exhibited increased free binding energy, elevated protein compactness, and a reduced number of atomic contacts, suggesting disrupted protein stability and interaction. Notably, our model revealed that the IR residues primarily stabilized the RfeB binding sites located in the central region (CR). This computational approach for protein mutagenesis could provide a foundation for future experimental studies on the functional characterization of RfeB and other homeodomain-containing proteins in *T. marneffei*.

## 1. Introduction

Transcription factors are proteins that specifically bind to DNA and modulate the expression of target genes; hence, they play essential roles in the regulation of numerous biological processes in living organisms. Aberrant activities of transcription proteins are often detected in various human diseases [1]. Also, transcription factors are critical drivers of virulent traits for human pathogens [2]. As multiple signaling pathways ultimately converge to transcription factors, it is proposed that therapeutic drugs targeting the transcription proteins can be more specific with fewer side effects than targeting the upstream components of the pathways [3]. One way to exploit transcription factors as therapeutic agents is to target specific protein structures necessary for protein-protein interaction and protein-DNA interaction. In silico approaches have become powerful tools in predicting 3D protein structures, protein-protein interaction, and small-molecule binding, which can assist large-scale rational drug design. The use of computers and bioinformatics technologies drastically reduces the time and the cost of animal model studies, which is beneficial for pharmaceutical research and drug discovery, especially in countries with low income and resources.

Over 100 years ago, the term homeosis was coined to refer to the biological process where something has been changed into the likeliness of something else [4,5]. Homeotic proteins (Hox) are transcription factors that control the homeotic transformation of the cells by means of regulating expressions of high-level ‘executive’ genes, such as other transcription factors and signaling molecules [6]. The Hox proteins contain a special sequence called the homeobox, which translates into a 60-amino acid homeodomain that normally functions as a DNA-binding domain. In animals, the *Hox* cluster genes control the body plan identity on the anterior–posterior axis [7]. In *Drosophila melanogaster*, for example, a change in the order of the *Hox* clusters leads to the homeotic transformation of the body plan, resulting in the famous antennapedia phenotype, where legs were found in place of the antenna [8,9]. In humans, mutations in *Hox* genes typically cause malformation in the anterior or posterior of the body segments and organs [10]. In plants, the homeobox transcription factors govern the developmental programs of flowers, leaves, and vegetative shoots [11]. For instance, in *Arabidopsis thaliana*, overexpression of the Hox protein *kn1* results in leaf morphology alteration [12]. Overall, the homeobox proteins function as master regulators that govern developmental processes in multiple organisms. Given their roles in developing multicellularity and cell differentiation, the homeobox genes have evolved and expanded to more than 100 genes for each species of animal and plant.

In the kingdom of fungi, the number of homeobox genes seems to be limited to under a dozen [13]. Strikingly, this transcription factor family plays a significant role in the regulation of sexual reproduction, morphogenesis, and metabolism across fungal members of Ascomycota and Basidiomycota (Table 1 and Table 2). For multicellular fungi, the homeobox transcription factors control fruiting body development. In the Ascomycete *Podospora anserina*, the *pah2* and *pah5* homeobox genes play essential roles in fruitification and sexual asci spore formation [14]. In the Basidiomycete fungi such as the mushrooms *Volvariella volvacea*, *Coprinopsis cinerea*, and *Schizophyllum commune*, the homeobox genes are differentially expressed in each developmental stage of fruiting body formation [15,16,17]. For unicellular yeast, the homeobox proteins control sexual development and invasive morphology. In the Ascomycete yeast *Saccharomyces cerevisiae*, the MATa1/alpha2 are the homeodomain proteins involved with cell type determination and sexual reproduction [18]. The Ste12 from *S. cerevisiae* regulates mating and invasive/pseudohyphal growth [19]. In the Basidiomycete yeast *Cryptococcus neoformans*, the Sxi2a and Sxi1α coordinately control sexual development [20]. Furthermore, the homeobox proteins participate in the regulation of diverse metabolic pathways. In the aflatoxin-producing fungus, *Aspergillus flavus*, the Hbx1 is necessary for aflatoxin production [21,22]. Likewise, the Hbx1/HbxA orthologous proteins govern a myriad of secondary metabolisms in *Aspergillus fumigatus* and *Aspergillus nidulans* [23,24,25]. In *Neurospora crassa*, the homeodomain kal-1 transcription factor regulates colony morphology formation, conidiation, and nutrient sensing [26]. The Grf10 from the human fungal pathogen *Candida albicans* is important for not only filamentation but also the response to adenine, copper, and phosphate [27,28,29].

In the case of fungal pathogens, the ability of a fungus to develop proper morphology during infection is profoundly linked to its virulence [74,75]. Given the strong conserved role of homeobox proteins in regulating morphology and infection-related structure, the homeobox proteins contribute to pathogenicity in many fungal pathogens such as *C. albicans*, *Ustilaginoidea virens*, *Magnaporthe oryzae*, *A. fumigatus*, and *Colletotrichum* spp. [23,27,31,32,34,41].

*Talaromyces marneffei* is an emerging opportunistic fungal pathogen that causes endemic mycoses in Southeast Asia. The fungus exhibits true filamentous growth at 25 °C and undergoes a dimorphic transition to unicellular fission yeast at 37 °C. Yeast cells are presented in infected patients, and therefore, yeast morphology is considered as the pathogenic form [76]. Infection usually occurs by inhaling conidia into the host’s lungs, suggesting that conidia are infectious agents [76]. In *T. marneffei*, the *stlA*, homolog of Ste12, is the only homeodomain-containing transcription factor that has been characterized in detail [52]. The *stlA* gene deletion mutant exhibited normal vegetative growth, asexual development, and dimorphic switching. The *stlA* gene from *T. marneffei* could complement the sexual defect of the *A. nidulans steA* mutant. Thus, the StlA protein likely has an implicated role in sexual production. Other homeobox proteins, however, are uncharacterized in this fungus.

In this current study, we identified all additional putative homeodomain-containing proteins and assessed their biological relevance by observing the gene expression levels of seven selected homeobox candidates in different cell types and during the morphological phase transition. To understand the structure of homeobox proteins, we focused on the selected protein RfeB and conducted an in silico prediction of RfeB-protein interaction. Our data revealed several key residues and binding sites that are important for protein complex formation, which could elucidate the cellular function of fungal homeobox proteins and might have applications for novel antifungal drug design.

## 2. Materials and Methods

### 2.1. Strains, Media, and Culture Conditions

*T. marneffei* ATCC20051 (CBS119456, F4) strains were cultured on Potato Dextrose Agar (PDA) at 25 °C for 10–14 days to generate conidia. Conidia were harvested from culture plates by scraping the surface of the colony and resuspending in sterile 1× Phosphate Buffered Saline with 0.1% *v*/*v* tween 40. The conidia suspension was filtered through Miracloth (Calbiochem, Germany), and a hematocytometer was used to determine conidia concentrations.

Preparation of *T. marneffei* in different growth forms was conducted as previously described [77]. Briefly, the conidia were harvested, and 1 × 10^8^ conidia/mL were inoculated in a 50-mL Sabouraud’s dextrose broth (SDB). Cultures were incubated at 25 °C (mycelium phase) and 37 °C (yeast phase) with continuous shaking at 200 rpm. After 72 h, the stable mycelium and yeast cultures were collected by centrifugation at 4 °C, 5800× *g* for 30 min. For the phase transition experiment, 1 × 10^8^ conidia/mL were inoculated in a 5-mL SDB and incubated at 25 °C (mycelium phase) and 37 °C (yeast phase) with continuous shaking at 200 rpm. Cultures were harvested at 30-min, 1-h, and 2-h by centrifugation at 4 °C, 5800× *g* for 30 min.

### 2.2. Quantitative Real-Time PCR

Total RNA was isolated by using TRIzol^®^ reagent (Thermo Fisher Scientific, Walthman, MA, USA), treated with DNase I, and converted to cDNA as previously described [78]. Quantitative real-time PCR was performed using the SYBR Green qPCR mix (Thunderbird SYBR Green Chemistry, Toyobo, Osaka, Japan). An actin gene was included as a reference gene (internal control) [79]. All primers used in this study are listed in Table 3. Calculation of a relative expression was performed using the 2^−(∆Ct)^ where ∆C_t_ = (C_t_ target − C_t_ actin). The Student’s *t*-test and statistical significance were calculated using Excel (Microsoft 365 MSO). Error bars indicate standard deviation, calculated using Excel.

### 2.3. Sequence Analysis

Homeobox transcription factors were identified using InterPro terms IPR009057 (homeodomain-like), IPR003120 (STE-like), and IPR001356/IPR001827 (homeobox) via the web-based Fungal Transcription Factor Database: http://ftfd.snu.ac.kr/ (accessed on 2 February 2023) [80]. For protein sequence comparison, RfeB and Grf10 protein sequences were retrieved from Uniprot and the Candida Genome Database [81], respectively, and aligned using the SIM tool: https://web.expasy.org/sim/ (accessed on 26 June 2024). A graphical representation of RfeB and Grf10 alignment was generated using the LALNVIEW program. Visualization of the *T. marneffei* Hox protein functional domains and key residues was performed using the IBS 2.0 tool: https://ibs.renlab.org (accessed on 27 January 2023) [82]. STRING [83] was used to predict the protein-protein interaction network: https://string-db.org/cgi/input.pl (accessed on 22 November 2023).

### 2.4. Heatmap Generation

A heat map indicating transcript abundance and expression pattern was constructed using the HEATMAP hierarchical clustering web tool: https://www.hiv.lanl.gov/content/sequence/HEATMAP/heatmap.html (accessed on 24 May 2023). The log_2_ fold-change values of relative expression levels (the 2^−(∆Ct)^ value) were used to generate the heatmap.

### 2.5. Phylogenetic Analysis

Amino acid sequences of 141 homeobox proteins from 17 diverse fungal species (Appendix A) were retrieved from: http://ftfd.snu.ac.kr/ (accessed on 2 February 2023), using InterPro term (IPR001356) [80]. The Clustal Omega program was employed to perform protein sequence alignment: https://www.ebi.ac.uk/Tools/msa/clustalo/ (accessed on 26 June 2024). A maximum likelihood tree was constructed in MEGA 11 with a bootstrap analysis of 100 replicates. The phylogenetic tree was further decorated using the web-based tool iTOL: Interactiv Tree of Life: https://itol.embl.de/ (accessed on 31 January 2023) [84].

### 2.6. Modeling Structure of RfeB in Complex with TmSwi5

The structure of RfeB in complex with TmSwi5 was predicted using AlphaFold3 [85], as available on: https://alphafoldserver.com (accessed on 5 September 2024), employing the protein sequences as demonstrated in (Appendix A). The double-strand DNA was added to the complex using the sequences following the previous study [86]. The structure of Swi5 contains three zinc-binding domains, and therefore, zinc ions (Zn) were added to the structure during the modeling. The protein-protein complex was modeled using the AlphaFold3 tool. The position of predicted residues was monitored based on the AlphaFold3 confidence scores (plDDT).

The triple mutation of RfeB was performed using in silico mutagenesis by replacing the residues Asp276, Glu279, and Gln282 using the protein editing tool in the BIOVIA Discovery Studio 2022 visualizer.

### 2.7. Molecular Dynamics Simulation

Molecular dynamics simulations were performed on an HPE Apollo 6500 (Hewlett Packard Enterprise, Spring, TX, USA) based on AMD EPYC 7742 with NVIDIA A100 SXM4 GPU (San Francisco, CA, USA). GROMACS 2022.4 package [87] was used to perform simulations. The protein-protein complex in the presence of double-strand DNA was prepared using AmberTools23 [88] with the tleap module. The structure of the protein was parameterized using the AMBER ff14SB force field [89]. The standard protonation state of titratable residues was corrected using the ProPka server at pH 7.4 [90]. The bonds between Zn ions and Swi5 were modeled using the Zinc AMBER force field (ZAFF) model following Peters et al. [91]. The DNA was parameterized using the AMBER OL15 force field. The protein complex was placed in a cubic box with a distance of 20 Å between the protein surface and the edge of the box. Water with the TIP3P model was added to the system, followed by sodium (Na^+^) and chloride (Cl^−^) ions for neutralization at a concentration of 0.15 M. The topology and coordinates of the protein-protein complex were converted to GROMACS formats using the ParmEd script.

The protein-protein complex underwent energy minimization using the steepest descent (SD) and conjugate gradient (CG) algorithms with an energy tolerance of 1 kJ/mol/nm. The complex was then equilibrated using the constant number of particles, volume, and temperature (NVT) ensemble and the constant number of particles, pressure, and temperature (NPT) ensemble for 1000 ps each. The long-range electrostatic and van der Waals interactions were calculated using the Particle Mesh Ewald (PME) and cut-off methods with a cut-off value of 12 Å. All bonds to hydrogen atoms were constrained using the LINear Constraint Solver (LINCS) algorithm. The production run was performed for 500 ns at a temperature of 310 K and a pressure of 1 atm.

The dynamics of the proteins were monitored by calculating the root-mean-square deviation (RMSD) using the gmx_rms command. The compactness of the protein-protein complex was monitored using the radius of gyration (Rg) value, which was calculated using the gmx_gyrate command. The number of atom contacts was monitored using gmx_mindist with a cut-off value of 6 Å. The number of hydrogen bonds was monitored using VMD 1.9.4a53 software [92] with the hydrogen bonding analysis module.

### 2.8. Protein-Protein Binding Free Energy

The binding energy between the protein-protein complex was calculated using molecular mechanics with generalized Born and surface area (MM-GBSA) with the gmx_MMPBSA package [93]. The internal and external dielectric constants were set to 1 and 78.5, respectively. The salt concentration and solvent probe radius were defined as 0.15 M and 1.4 Å, respectively. The results were visualized using the gmx_MMPBSA_ana module as implemented in the gmx_MMPBSA package. The relative binding free energy was calculated using the following equation.
$$∆G_{\mathrm{bind}}=∆E_{\mathrm{MM}}-T∆S+∆G_{\mathrm{sol}}$$
$$∆E_{\mathrm{MM}}=∆E_{\mathrm{vdw}}+∆E_{\mathrm{elect}}$$
$$∆G_{\mathrm{sol}}=∆G_{\mathrm{polar}}+∆G_{\mathrm{nonpolar}}$$

## 3. Results

### 3.1. Homeobox Proteins in T. marneffei

The FTFD pipeline was used to identify putative transcription factors in *T. marneffei*. In total, there were 713 transcription factors, which could be classified into 43 protein families (Figure 1). The homeodomain-containing proteins comprised the second-largest family (Figure 1). A total of 119 homeobox transcription factors shared conserved homeodomain-like (IPR009057), STE-like (IPR003120), or homeobox/homeodomain (IPR001356 and IPR001827).

### 3.2. Phylogenetic Analysis of T. marneffei Homeodomain-Containing Proteins

To assess the evolutionary relationship of fungal homeobox proteins, the phylogenetic tree was constructed based on amino acid sequences of 141 homeobox proteins from 17 diverse fungal species (Appendix A). We examined ten homeobox proteins in *T. marneffei* genome that contain homeodomain. Phylogenetic analysis showed that homeobox genes were classified into seven separate groups, and *T. marneffei* homeobox proteins were clustered into six groups (Figure 2). The StlA protein (EEA24489.1) from *T. marneffei* was classified with other Ste-like proteins such as Ste12 from *S. cerevisiae*, SteA from *N. crassa* and SteA from *A. fumigatus.* The *T. marneffei* RfeB protein was clustered with FoHox11 from *F. oxysporum*, Pah1 from *P. anserine*, MoHox1 from *M. oryzae*, kal-1 from *N. crassa*, Afu4g10220 (RfeB) from *A. fumigatus*, and Pho2 from *S. cerevisiae*. Indeed, the RfeB protein family from *Aspergillus* spp., Hoy1 from *Yallowia lipolytica*, Pah1, MoHox1, and kal-1 were previously reported as the homologs of Pho2 from *S. cerevisiae* [94]. Together, these findings suggested that *T. marneffei* homeobox proteins in each subgroup were closely related to those of the corresponding fungal homeobox proteins.

### 3.3. Gene Expression Analysis of TmHox Genes

For further analysis, we focused on seven uncharacterized genes that contain the homeobox domain IPR001356, designated as TmHox1–TmHox7 (Table 4, Figure 3). We excluded the homeobox EEA26628.1 and EEA28535.1 proteins because they contain only the conserved site of the homeobox protein, antennapedia type (IPR001827) while the homeobox protein domain IPR001356 is absent (Appendix A). Also, the *StlA* gene was previously characterized to be expressed primarily in vegetative tissues (Table 5 [52,95]), and therefore excluded from our current study. Among the seven homeobox proteins, two proteins, TmHox2 and TmHox7, have a C_2_H_2_ zinc finger domain at the C-termini in addition to the homeobox domain (Figure 3). The TmHox4 was the only protein that contains the homeobox domain at the C-terminus (Figure 3).

To investigate the biological relevance of putative *TmHox* genes in vitro, we examined the transcript levels of *TmHox* genes from *T. marneffei* ATCC200051 strain growing in three different conditions: on PDA plates for conidiation (asexual development), in liquid SDB for hyphal growth at 25 °C, and yeast growth at 37 °C. Gene expression patterns of seven *TmHox* genes were classified into three different groups. Group I was comprised of the conidia specific gene *TmHox4* (Figure 4A). The *TmHox4* transcripts were highly accumulated during conidiation, being 500-fold higher than transcript levels presented in the hyphal or yeast phases. This result is consistent with DNA microarray study by Pasricha et al., 2013, showing that *TmHox4* gene homolog from *T. marneffei* ATCC18224 strain was accumulated at high levels in asexual conidial cell (Table 5 [96]). Group II was classified as genes whose expression levels were strongly downregulated in conidia, including *TmHox1*, *TmHox5* (*RfeB*), and *TmHox7* (Figure 4B). Upregulated levels of *TmHox1* gene in stable mycelial form relative to stable yeast form have been detected in previous transcriptomic study (Table 5; *TmHox1* [95]). Also, the *RfeB* gene is highly expressed in both stable mycelial and yeast form, which is in agreement with RNA-sequencing data by Yang et al., 2014 [95]. Lastly, group III consisted of genes that were expressed at low levels in the yeast phase and were differentially upregulated in the mycelium phase, including *TmHox2*, *TmHox3* and *TmHox6* (Figure 4C). This expression pattern of *TmHox3* and *TmHox6* genes has been previously reported from different *T. marneffei* strains (*TmHox3* [97]; *TmHox6* [95,97]).

Gene expression analysis of hyphal, yeast and conidiation cell states only reflect gene expression patterns in late morphogenesis and development, which do not necessarily illustrate the transcriptional events during the cell-state transition [96]. To examine the gene expression profiles of TmHox genes in early phase transition, we measured the transcript levels of TmHox genes when conidia were exposed to temperature-induced morphological switching at 0.5-h, 1-h, and 2-h. The hierarchical heatmap was generated, using Log_2_ values of normalized gene expression levels (Figure 5). As expected, the *Hox* gene expression profile from mycelium and yeast phase transition formed a separate cluster, suggesting that *Hox* genes were transcriptionally regulated differently when transitioning to different morphologies (Figure 5A). *TmHox4*, *TmHox2*, and *RfeB* genes were greatly upregulated during the mycelium phase transition. For instance, *TmHox2* gene was significantly upregulated 7-fold when transitioning from conidia to mycelium phase (25 °C, 2-h) (Figure 5B). The study by Yang et al., 2014 [95] found that *TmHox2* gene was highly expressed during yeast to mycelium phase switching (Table 5 [95]). Additionally, our result on *TmHox4* gene is consistent with data by Pasricha et al., 2013, demonstrating that *TmHox4* gene was upregulated during phase transition from yeast (37 °C) to hyphal (25 °C) phases (Table 5 [96]). In contrast, *TmHox6* transcript was highly accumulated in conidia and its transcript levels were decreased during the early yeast and mycelium phase transitions (Figure 5C). These transcript profiles suggested that *TmHox4*, *TmHox*2 and *RfeB* could be involved in hyphal growth during the early stage. Altogether, our results suggested that homeobox genes were differentially expressed during the morphological development of *T. marneffei*.

### 3.4. Sequence Analysis of RfeB Protein in T. marneffei

The fungal Hox proteins were classified into 12 groups based on Vonk and Ohm analysis [98]. The Pho2 group represents one of the protein classes whose members have been well characterized in detail from diverse fungal species, including unicellular yeast vs. filamentous fungi and saprophytic vs. pathogenic fungi. Importantly, Pho2 and its homologous proteins play a prominent role in the regulation of morphological development and metabolic pathways [94]. In addition to the homeodomain, Pho2 contains the unique IR region, which is absent in other groups of fungal homeobox protein. In *S. cerevisiae*, the IR is necessary for Pho2 interaction with its three distinct protein partners, Bas1, Pho4, and Swi5 [37], to regulate the expression of genes in purine biosynthesis, phosphate limitation, and mating type switching pathways, respectively. Mutational analysis reveals that the CDDF-E residues form the core of the IR region [37]. In *T. marneffei*, the IR CDDF-E residues are found only in the RfeB protein (Figure 6A). Thus, the RfeB protein is a member of the Pho2 protein family based on the phylogenetic tree analysis and its conserved IR residues. We compared RfeB with Grf10 from the pathogenic yeast *C. albicans* using the SIM tool. As shown in Figure 7A, RfeB is composed of 575 amino acids, and there are three conserved regions: the homeodomain DNA-binding domain, the central region (CR) located between amino acids 150–245, and the IR region located between amino acids 250–325.

Since the CDDF-E residues are well conserved in RfeB, we hypothesized that RfeB could interact with other proteins to regulate specific downstream target genes. We performed a BLAST homology search against known Pho2 protein partners from *S. cerevisiae*, and found that *T. marneffei* showed the highest percentage identity with ScSwi5 (TmSwi5/PMAA_023500/EEA27475.1: 54.55%) while exhibited less conservation with Bas1 (Eta2/PMAA_062970EEA25176.1: 33.54%) and Pho4 (PalcA/PMAA_051940/ EEA21384.1: 31.48%). To further predict the RfeB interaction network, the amino acid sequences of RfeB were subjected to STRING analysis (Figure 6B). The RfeD (PMAA_029350) and TmHox4 (PMAA_096860) transcription factors showed the highest predicted score for being a protein partner of RfeB (Figure 6B).

### 3.5. In Silico Structure Prediction of RfeB-TmSwi5-DNA Complex

To evaluate how the conserved IR region contributes to protein complex formation, in silico mutagenesis of IR residues and molecular dynamics simulation were performed.

In the initial analysis, TmSwi5 was selected as the RfeB protein partner because TmSwi5 showed the highest homology with ScSwi5. To serve as a control and validate the accuracy of computational simulation, the known interaction of Pho2-ScSwi5-DNA was subjected to in silico structure and interaction prediction (Appendix A). The AlphaFold3 model predicted that Pho2 interacted with the DNA sequence of 5′-CAATTTA-3′ and ScSwi5 interacted with a sequence of 5′-AAACCAGCAT-3′ of *HO* promoter (Appendix A), corresponding to the previous results from methylation and hydroxy radical interference [99]. Next, in silico mutagenesis was conducted to replace the Pho2 conserved IR residues Asp371, Glu374, and Gln377 with alanine residues, generating the Pho2 mutated protein (Appendix A). Molecular dynamics of ScSwi5 with wild-type Pho2 and mutated Pho2 were simulated for 500 ns. The root-mean-square deviation (RMSD) and radius of gyration (Rg) were monitored to measure the dynamics and compactness of the protein complex. The results of molecular dynamics simulations are demonstrated in Figure 8. All complexes reached equilibrium after 100 ns of simulation. While the RMSD profile of Pho2 showed similar deviation in both the wild-type and mutant system (Figure 8A), the mutated Pho2-ScSwi5-DNA complex showed an increase in Rg values compared to the wild-type complex (Figure 8B), indicating the less compact structure. As expected, the alanine substitutions at the conserved residues Asp371, Glu374, and Gln377 resulted in a decrease in the relative binding free energy and atomic contacts with ScSwi5 (Table 6 and Figure 8C), highlighting their contribution to protein-protein interactions. Overall, this result suggested that the conserved IR residues contributed to Pho2-ScSwi5 protein interaction as previously verified in *S. cerevisiae* [37]. Thus, AlphaFold3 and simulation systems could accurately predict protein-protein complex structure with high confidence.

The structure of RfeB in complex with TmSwi5 and DNA was modeled using AlphaFold3. As DNA sequences or promotors regulated by RfeB protein are unknown in *T. marneffei*, we selected DNA sequences used for crystallization of the homeodomain Engrailed protein -DNA complex to incorporate into our simulation analysis [86]. Indeed, the co-crystal structure of Engrailed protein has been successfully used as a model to predict Pho2 residues that contact DNA and contribute to transcriptional activation [100]. The confidence of the position of each amino acid residue in the predicted structure was validated using the plDDT (Appendix A). The plDDT plot showed high confidence (plDDT > 90) in the DNA binding region of both proteins. The structure showed less confidence in other locations, especially the central region of RfeB (Appendix A). However, AlphaFold3 was unable to predict the full-length structure of these proteins (Appendix A). Some residues were removed, and only predictable residues were kept. Removal of unpredictable structures did not affect the overall model. The final model of this protein complex is demonstrated in Appendix A.

### 3.6. Molecular Dynamics Simulation

Molecular dynamics simulations were carried out to study the complex stability and dynamics of the protein-protein complex under physiological conditions. To test if the conserved IR residues Asp276, Glu279, and Gln282 contribute to RfeB protein interaction, we conducted in silico mutagenesis by substituting the residues Asp276, Glu279, and Gln282 with the alanine, generating the triple RfeB mutated protein. Then, the protein-protein complex of TmSwi5 and DNA with the wild type or the triple RfeB mutated proteins was simulated for 500 ns.

RMSD profiles (Figure 9A) of both the wild-type and triple-mutated proteins showed similar deviations. TmSwi5 from both the wild type and mutated RfeB models reached equilibrium after 300 ns of simulation and remained stable thereafter. In the case of RfeB, both the wild type and mutated proteins reached equilibrium after 20 ns of simulation. Both wild-type RfeB and TmSwi5 demonstrated high RMSD values, indicating significant conformational changes during the simulation compared to the initial structure. This result indicated that the wild-type RfeB-TmSwi5 complexes underwent stable structural changes. However, the model with mutated RfeB protein showed an unstable protein-protein complex as the RfeB unbounded from the TmSwi5 after 50 ns of simulation (Figure 9D). This evidence suggests that the residues Asp276, Glu279, and Gln282 play an important role in the binding between RfeB and the TmSwi5.

When monitoring the Rg values (Figure 9B), the wild-type RfeB-TmSwi5-DNA complex showed a lower Rg value after simulation, indicating that the protein-protein complex adopted a more compact structure. In contrast, the mutated RfeB-TmSwi5-DNA complex demonstrated a higher Rg value than the wild-type complex. This high value did not change during the simulation, suggesting that the protein-protein complex was more extended, resulting in a larger overall size.

To gain insight into the protein-protein interactions between RfeB and TmSwi5, the number of atomic contacts between these proteins was monitored, as depicted in Figure 9C. The wild-type RfeB showed a greater number of atomic contacts with TmSwi5 than the mutated RfeB, indicating greater protein-protein interactions and complex stability. Overall, the molecular dynamics simulations of both the wild-type and mutant RfeB-TmSwi5 complexes converged in 500 ns of simulations.

### 3.7. Protein-Protein Binding Free Energy

To explore the binding ability between RfeB variants and TmSwi5, MM-GBSA calculations were employed to quantify the relative binding free energy, as demonstrated in Table 6. The wild-type RfeB showed 3-fold greater binding energy than the mutated RfeB, being −96.98 ± 11.86 in wild-type vs. −35.22 ± 9.55 kcal/mol in mutated proteins. The wild-type RfeB exhibited stronger van der Waals interactions than the mutated RfeB, suggesting a more stable complex and stronger protein-protein interactions. In the case of electrostatic interactions, the mutated RfeB showed less favored electrostatic interactions than the wild-type RfeB. This may be due to the unbinding of mutated RfeB from the TmSwi5 surface. Additionally, the polar solvation was more negative in the mutated RfeB, suggesting that the solvent molecules favorably stabilized the electrostatic interactions due to the greater solvent-exposed surface in the central region. The polar solvation was consistent with the electrostatic interaction calculated in the gas phase. In the case of non-polar solvation, the mutated RfeB protein tended to have slightly fewer negative values than the wild-type RfeB, suggesting a slight reduction in favorable non-polar interactions with solvent molecules. Overall, the binding free energy of mutated RfeB is less negative than that of the wild type, demonstrating a less stable protein-protein complex. Thus, the mutations at conserved residues Asp276, Glu279, and Gln282 destabilized the protein-protein complex and overall interactions.

To further explore how specific residues contribute to the overall binding energy between each protein-protein complex, per-residue energy decomposition was calculated (Figure 10). The per-residue energy decomposition revealed three major binding sites between wild-type RfeB and TmSwi5, as depicted in Figure 10. Importantly, the RfeB key interacting residues (Figure 10C, highlighted in gray for binding energy less than −1 kcal/mol) were Arg45, Leu49, Gln51, Gln52, Gln53, Asn55, Gln56, Ala61, Thr62, Glu85, Asn92, Thr94, His141, Arg146, Glu147, Asn267, Ser268, and Phe272. In the case of TmSwi5, the key binding residues were Thr381, Phe384, Ser385, Pro386, Arg390, Leu391, Pro393, Thr414, His459, Leu460, Asp462, Arg463, Asn499, Val500, Phe501, Ala502, Met563, and Ser564 (Figure 10C).

Notably, the first binding site was located at the homeodomain of RfeB and the DNA binding domain of TmSwi5 (Figure 10A,B). At this site, RfeB residues 42 to 95 interacted with TmSwi5 residues 381 to 393 and residues 497 to 502. As this site provided the most decomposition energy to the protein complex, the binding site 1 was the major protein-protein interaction region. The second binding site was composed of the RfeB residues 141 to 147 and the TmSwi5 residues 561 to 564 (Figure 10A,B). The residues 267 to 272 located at the central region of RfeB, and residues 414 to 463 of Swi5 formed the third binding site. However, both the second and third binding sites provided slight protein-protein binding ability.

Surprisingly, the per-residue energy profile indicated that the conserved IR residues Asp276, Glu279, and Gln282 did not directly bind to TmSwi5. Yet, alanine substitutions within these residues resulted in the loss of the protein-protein interactions at binding sites 2 and 3 (Figure 10C), highlighting the importance of these residues during protein complex formation. In fact, these conserved IR residues stabilized the secondary structure of the RfeB central region of RfeB (i.e., binding sites 2 and 3), making the overall protein structure suitable for binding with TmSwi5. As depicted in Figure 10A,B, the residue Asp276 is located at the loop region between the α-helix and β-sheet structures, forming an intramolecular hydrogen bond with Ser284, with high occupancy (Figure 10B) and stabilizing the secondary structure at the central region. This Asp276—Ser284 hydrogen bond led to another internal formation of hydrogen bonds between Glu279—Trp186/Arg188, as well as Gln282—Ser185. These crucial residues are located near binding site 3 of RfeB. Mutations at these residues resulted in the loss of proper RfeB protein folding between α-helix and β-sheet structures, leading to a decrease in binding ability with TmSwi5, especially at binding site 3. The superimposition between the wild-type and mutant RfeB obtained from molecular dynamics simulations showed substantial alteration in the secondary structure of the RfeB central region (Appendix A).

Consistently, the Pho2 residue Asp371 was also found to stabilize the α-helix and β-sheet structures by forming intramolecular hydrogen bonds with Asn379 with high occupancy (Appendix A). Also, the Pho2 residue Gln377 interacted with Ser206 located in the central region to stabilize the Pho2 protein tertiary structure (Appendix A). Thus, the internal formation of hydrogen bonds by the IR-conserved aspartate and glutamine residues was observed in both RfeB and Pho2 protein complexes. Putative functional domains of RfeB and the contribution of IR residues to RfeB protein stability and interaction are summarized in Figure 7B.

## 4. Discussion

Transcription factors were historically considered “undruggable” due to the limitation on structural information, especially about the ligand-binding pockets and protein-protein interaction [3]. Currently, advances in machine-learning algorithms and bioinformatics analytic tools have accelerated the prediction of 3D protein structure and interaction with proteins or small molecules, which can ultimately guide the future design of drugs targeting transcription factors. For example, the compound MCULE-7146940834 is a potential novel antimalarial drug candidate as it is predicted in silico to bind at the active site of *Plasmodium falciparum* AP2-I transcription factor, a key regulator of red blood cell invasion during human infection [101]. Belzutifan is an FDA-approved anti-cancer medication that inhibits the HIF-2a signaling pathway by disrupting the protein-protein interaction between the HIF-2a transcription factor and the ARNT protein partner [102]. A comprehensive list of small molecules that can modulate transcription factors and are approved or undergoing clinical development for human diseases has been reviewed [3]. Also, the transcription factors with antifungal potentials are well summarized in this review [103]. Thus, a previous “Herculean task ” for targeting transcription factors has recently become more amenable for drug discovery and therapy.

The family of homeodomain-containing transcription factors represents a promising unconventional target for developing therapeutic agents and biomarkers. This protein family regulates essential biological processes, and the expression of homeobox genes is oftentimes altered in normal vs pathogenic states [104,105]. In fungi, the functions of homeobox proteins are mostly associated with cellular and developmental processes. In the case of fungal pathogens, the homeobox proteins significantly contribute to virulence (Table 1 and Table 2). In this study, the genome-wide analysis of homeobox genes in *T. marneffei* was performed, and the in vitro biological function was assessed based on their gene expression profiles during morphological development. Bioinformatics analyses revealed that *T. marneffei* contained ten homeodomain-containing transcription factors. In general, the number of fungal homeobox genes is in the range of 12 [7,13]. Overall, the number of homeobox genes found in *T. marneffei* agrees with the diversification of homeodomain-containing proteins found in other fungi [34].

*T. marneffei* can exist in three cell types, each with different morphologies and functions: conidial cell is the infectious agent, mold is the environmental form, and yeast is the pathogenic form. In stable morphologies, the transcript levels of all tested Hox genes were altered when *T. marneffei* was grown in different cell fates. The TmHox4 transcripts were highly accumulated during conidiation. On the other hand, the transcripts of *TmHox1*, *TmHox5* (*RfeB*), and *TmHox7* were accumulated the least in conidia. Lastly, *TmHox2*, *TmHox3,* and *TmHox*6 showed differential gene expression between mycelium and yeast forms. During the early phase transition, we found that Hox genes were under transcriptional upregulation at higher degrees when switching from conidia to mycelium than when switching from conidia to yeast. The *TmHox4*, *TmHox2*, and *TmHox5* (*RfeB*) genes were among those Hox genes that increased their transcript levels during the transient mycelium phase transition. As mold is a multicellular fungus, this is consistent with the common role of homeobox genes in multicellular development found in other true molds, mushrooms, and eukaryotes [98]. In fact, the homeodomain proteins have been proposed to be the ancestral molecular toolkit that allows some of the eukaryotes to achieve multicellularity and complex organization [106]. Interestingly, this specific transcription factor family was completely lost in several lineages of unicellular parasites and intracellular symbionts [13]. For the pathogenic phase, *T. marneffei* is classified as a facultative intracellular pathogen, residing inside macrophages and tissue histiocytes as the parasitic unicellular yeast form [107]. We speculated that less transcriptional regulation of Hox genes during the transition from conidia to yeast phase agrees with a less prominent role of homeodomain-containing transcription factors in unicellular and intracellular lifestyles.

Phylogenetic analysis of 141 homeobox proteins from 17 diverse fungal species revealed that *T. marneffei* homeobox proteins were clustered into six groups. However, the homeobox proteins that clustered in the same group did not exhibit a similar gene expression profile. One possible explanation is that the homeobox transcription factors were grouped together based on the similarity mostly in the homeodomain. However, the rest of the full-length proteins may contain different sequences or domains that can have other specific functions. The discrepancy in Hox gene classification based on sequence similarity and gene expression pattern has been observed in several fungi, such as straw mushroom *V. volvacea* and plant pathogens *Fusarium* spp. [15,108,109]. Thus, the phylogenetic tree analysis, in combination with gene expression profiling, could provide a deeper insight into the comprehensive functional characteristics of this transcription factor family in *T. marneffei*.

The IR region is the signature sequence of Pho2 family proteins and exemplifies the functional region outside the homeodomain. In *S. cerevisiae*, Pho2 activates the transcription of its target genes in a combinatorial manner [110,111]. The IR region located within the 343–390 of Pho2 protein is proposed to form an interactive surface for protein-protein binding [37]. The CDDF-E residues of the IR region (Figure 6A) are proposed to form the core of the interaction region since point mutations in these residues disrupt the expression of all Pho2 target genes [37]. In *C. albicans*, the critical residues within the core of the IR region are necessary for Grf10 to regulate filamentation, adenine starvation, and copper toxicity response [29,94]. These results are consistent with the model that the IR region mediates protein-protein interaction, allowing Grf10 in *C. albicans* to activate different sets of target genes with different co-regulators. Despite the high conservation through fungal diversification, the role of IR residues has not been investigated in other fungi outside the yeast *S. cerevisiae* and *C. albicans*. Moreover, the 3D structure has never been characterized in Pho2 or any Pho2 homologs. In this study, an in silico approach was carried out to model the protein-protein interaction of Pho2 homolog, RfeB. Molecular dynamics simulation showed that RfeB contains 3 binding sites. The main binding site (binding site 1) was located within the homeodomain of RfeB. The other two binding sites were in the central portion, overlapping with the conserved CR and IR regions (Figure 7B). The RfeB residues that potentially mediate protein-protein interaction include Ser268, which showed the highest percentage of hydrogen bond occupancy with TmSwi5 (Figure 10C), and Phe272, which showed high sequence conservation in many fungal species (Figure 6A and Figure 10C). These potentially identified residues will be important candidates for mutagenesis studies in the future.

Based on our model, the conserved IR residues Asp276, Glu279, and Gln282 did not make any contacts with partner candidate TmSwi5 but rather internally stabilized the secondary structures of binding site 2 and binding site 3, allowing RfeB to fully interact with TmSwi5. Molecular dynamics simulation of wild-type proteins suggested that the IR conserved aspartate residue, i.e., Asp276 in RfeB and Asp371 in Pho2, is the most important residue among the three selected residues. Our model demonstrated that this specific aspartate residue stabilized the secondary structure of α-helix and the loop region by forming a hydrogen bond with the next eight- residues, i.e., Asp276 with Ser284 for RfeB and Asp371 with Asn379 for Pho2 (Figure 10A and Appendix A). Additionally, these aspartate-dependent interactions showed the highest percentage of hydrogen bond occupancy in comparison to the other two conserved residues for both RfeB and Pho2 models (Figure 10B and Appendix A). In supporting this function, in silico alanine substitution at Asp276, Glu279, and Gln282 residues of RfeB destabilized the proper folding of binding site 2 and binding site 3, leading to the loss of stable protein-protein interaction between RfeB and TmSwi5. Notably, the binding site 1 remained in contact with TmSwi5 in the model with mutated RfeB. However, this binding site alone was not sufficient for full interaction with TmSwi5. This is consistent with the previous in vivo experiments in Pho2, demonstrating that the homeodomain alone is not sufficient for cooperative DNA binding with ScSwi5 [99]. Thus, the CDDF-E residues of RfeB from *T. marneffei* were important for stabilizing the RfeB interaction sites, especially the binding sites 2 and 3. This new function of the conserved IR residues is still consistent with the studies in *S. cerevisiae* and *C. albicans*, which show that mutations within these residues disrupt the functions of Pho2/Grf10 in all pathways. Alternatively, the CDDF-E residues can still directly interact with other protein partners that we have not tested in this current study.

Our study revealed the role of the RfeB central portion in proper protein folding, protein stabilization, and interaction. This middle part of RfeB exhibits homology with the so-called “central region” in Grf10 from *C. albicans* (Figure 7A) and Pho2 from *S. cerevisiae*. Brazas et al. show that amino acid substitutions or protein truncation within this central region of Pho2 strongly affect the protein levels in *E. coli* [111] and *S. cerevisiae* [37], suggesting that this region contributes to proper protein folding and stability. For RfeB, the residues Ser185, Trp186, and Arg188 form hydrogen bonds with the IR-conserved residues (Figure 10A,B) to stabilize the protein structure, enabling RfeB to fully interact with Swi5. For Pho2, residue Ser206, located in the central region, also forms hydrogen bonds with IR residue. Thus, our simulation model agrees with the role of this central region in proper protein folding, stability, and protein-protein interaction. All key residues and regions identified from our in silico approach are summarized in Figure 7B.

TmSwi5 is one of the potential protein partners of RfeB in *T. marneffei*. The *HO* gene encodes the homothallic switching endonuclease, requiring gene conversion at the mating locus by generating a double DNA break. However, the *HO* gene shows no homology in the *T. marneffei* genome. This result suggests that RfeB may interact with TmSwi5 to control different targeted pathways, or it may work with different protein partners to regulate the same pathways (i.e., purine and phosphate starvation response pathways). In filamentous fungus *A. fumigatus*, Ace2, the Swi5 homolog, controls pigment production, conidiation, cell wall architecture, and virulence [112]. In pathogenic yeasts *C. albicans*, Ace2 regulates morphogenesis through the Regulation of Ace2 and Morphogenesis (RAM) pathway, growth under hypoxia, cell separation, adherence, biofilm formation, and virulence [113,114,115]. In addition to TmSwi5, the STRING interaction network analysis of RfeB showed enrichment in DNA-binding transcription factor activity, which is consistent with the prediction that RfeB likely interacts with other transcription factors to regulate target genes. It would be of great interest in the future to experimentally characterize the role of RfeB, the functions of the IR region, and the RfeB protein partners in true mold as well as dimorphic fungi.

## 5. Conclusions

In summary, we identified the genes encoding for homeodomain-containing proteins in the dimorphic fungal pathogen *T. marneffei*. Based on gene expression analysis, the differentially expressed homeobox genes suggested that they likely have biological functions in vitro, especially the regulation of thermal-induced dimorphism in *T. marneffei*. Based on our simulation model, the conserved IR residues contribute to the RfeB-protein interaction by stabilizing the secondary structures of binding sites. Identifying the function of conserved IR residues and other regions within the Pho2 protein family offers unique future opportunities for pharmaceutical research and medical intervention. For instance, broad-spectrum antifungal agents may be developed by characterizing small molecules that can specifically bind within this IR region and modulate pathogenic traits. Further experimental validations should be performed to provide insights into the functions and mechanisms of this specific transcription factor family.

## Figures and Tables

**Figure 1 jof-10-00687-f001:**
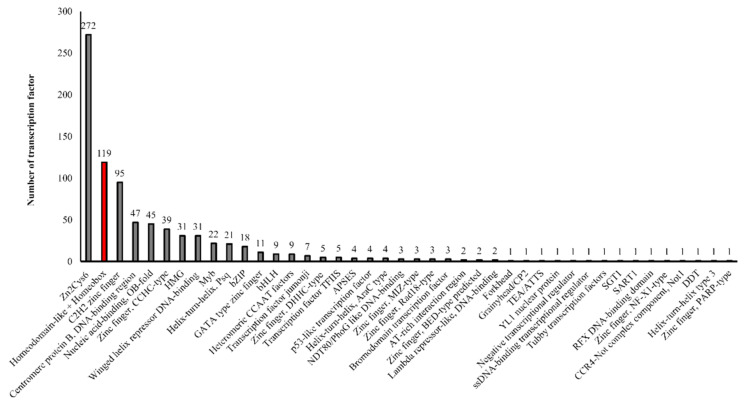
Transcription factor families in *T. marneffei* are depicted. 713 transcription factors are distributed in 43 families. The number of genes in each family are indicated on the top of each column. Homeodomain-like and Homeobox family is highlighted in red bar.

**Figure 2 jof-10-00687-f002:**
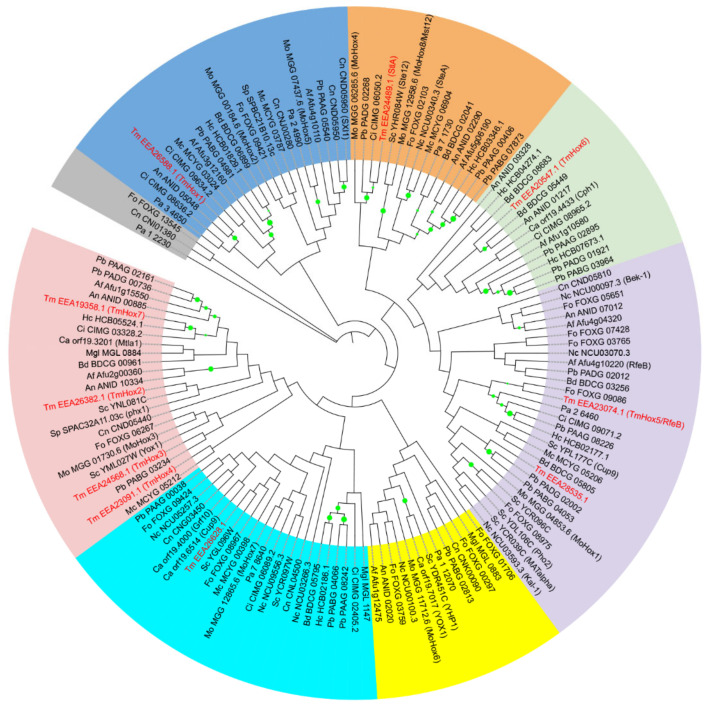
The phylogenetic tree of 141 putative and characterized homeobox transcription factors in fungi. Homeobox proteins were identified using InterPro term (IPR001356) via the pipeline of Fungal Transcription Factor Database (http://ftfd.snu.ac.kr/, accessed on 24 May 2023). The amino acid sequences of representative fungal homeodomain-containing proteins were aligned by the ClustalW tool. A maximum likelihood tree was constructed by MEGA11 software and decorated by iTOL program (https://itol.embl.de/, accessed on 24 May 2023). Bootstrap analysis of 100 replicates was performed and values greater than 40% are shown with a green dot. The putative *T. marneffei* homeobox proteins are highlighted in red.

**Figure 3 jof-10-00687-f003:**
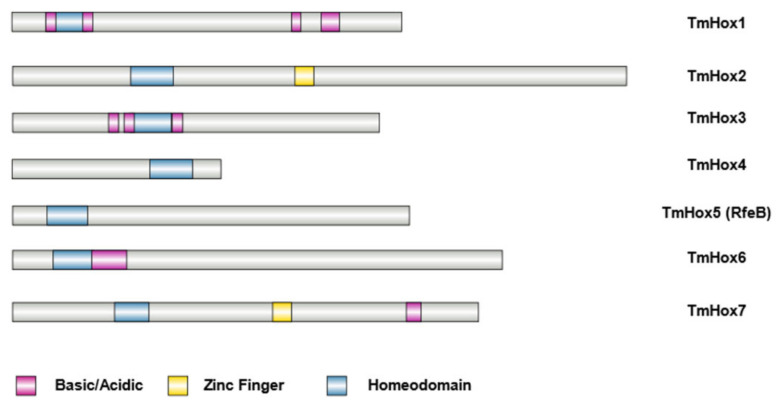
Bioinformatics analysis of the putative homeobox proteins in *T. marneffei.* Domain architecture of the putative homeobox proteins in *T. marneffei* is illustrated. Sequence features of homeobox proteins include Homeodomain, Zinc Finger, and Basic/Acidic regions. Protein domain architectures were generated using the IBS web-based tool.

**Figure 4 jof-10-00687-f004:**
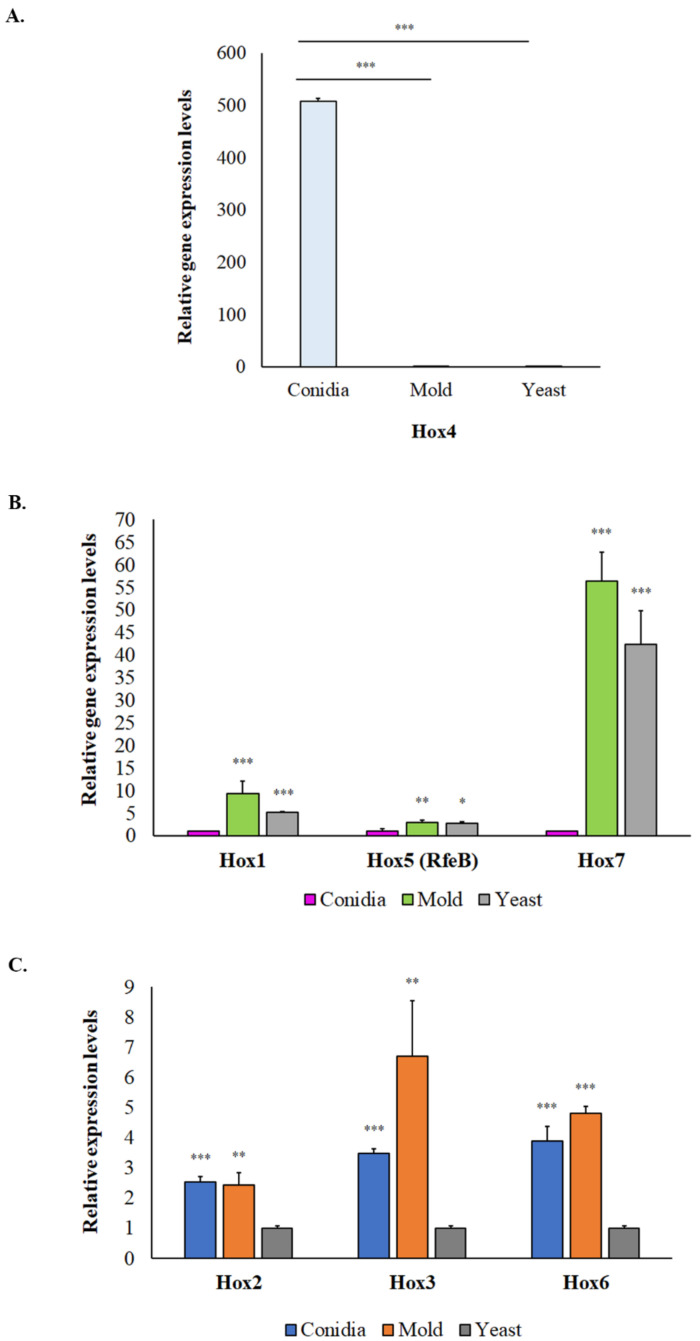
Hox genes are differentially expressed in *T. marneffei* growing in different cell states. Gene expression levels were assessed by qRT-PCR when *T. marneffei* was grown in different morphologies. Conidia of *T. marneffei* strain ATCC20051 were inoculated into SDB media and incubated at 25 °C (mycelium) and 37 °C (yeast). The 72-h cultures and conidia were harvested, and RNA was prepared (see materials and methods for details). Relative gene expression levels were calculated by the 2^−∆Ct^ method using actin as a reference gene. Fold-change levels were normalized to 1 for the phase where the transcript was the lowest. Gene expression profiles were classified into three groups. (**A**) Group I was classified as Hox genes with high accumulated levels in conidia or as conidia specific Hox genes. (**B**) Group II was classified as Hox genes with low expression levels in conidia. (**C**) Group III was classified as Hox genes with low expression levels in yeast and exhibited differential expression between stable mycelium and stable yeast. Experiments were performed in three biological replicates. Error bars indicate standard deviation. Statistically significant values (* *p* ≤ 0.05, ** *p* ≤ 0.01, *** *p* ≤ 0.001) are indicated.

**Figure 5 jof-10-00687-f005:**
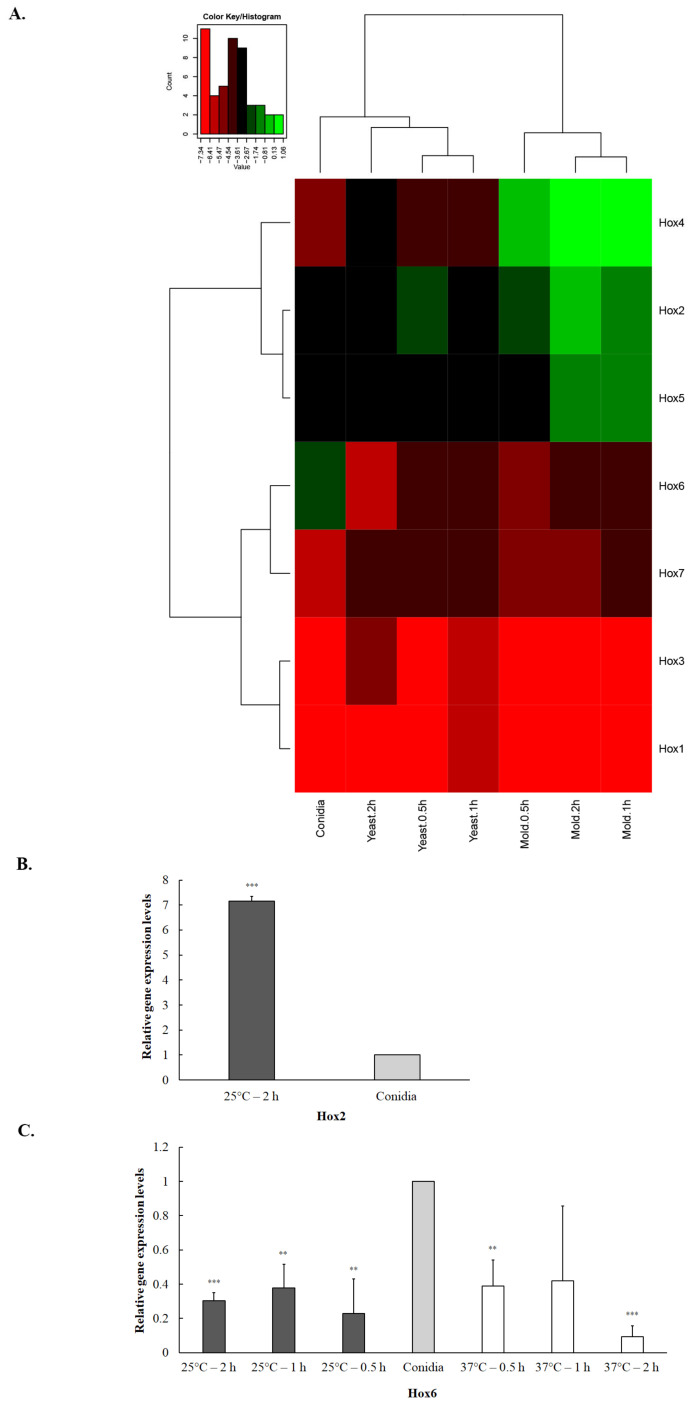
Hox gene expression patterns during early phase transition are depicted. Hox gene expression profiles were assessed by qRT-PCR when *T. marneffei* underwent phase transition from conidia to mycelium and yeast phases. Conidia of *T. marneffei* strain ATCC20051 were inoculated into SDB media and grown at 25 °C (mycelium) and 37 °C (yeast). Samples were harvested at 0.5-h, 1-h, and 2-h and subjected to RNA extraction, cDNA synthesis, and qRT-PCR (see materials and methods for details). Relative gene expression levels were calculated using the 2^−∆Ct^ method, using actin as a reference gene. (**A**) Heatmap illustrates the gene expression profiles of Hox genes during early phase transition. The names of the homeodomain protein-encoding genes are provided on the right side, and growth conditions are provided at the bottom. Dendograms are on the left and on top of the heatmap. Heatmap was generated using the log_2_ of relative gene expression values (2^−∆Ct^). Description of different colors on heatmap are depicted at the left upper corner. Gene expression levels of Hox2 (**B**) and Hox6 (**C**) are shown. Fold-change levels of conidial cells were normalized to 1. Experiments were performed in three biological replicates. Error bars indicate standard deviation. Statistically significant values (** *p* ≤ 0.01, *** *p* ≤ 0.001) are indicated in comparison to gene expression levels in conidia.

**Figure 6 jof-10-00687-f006:**
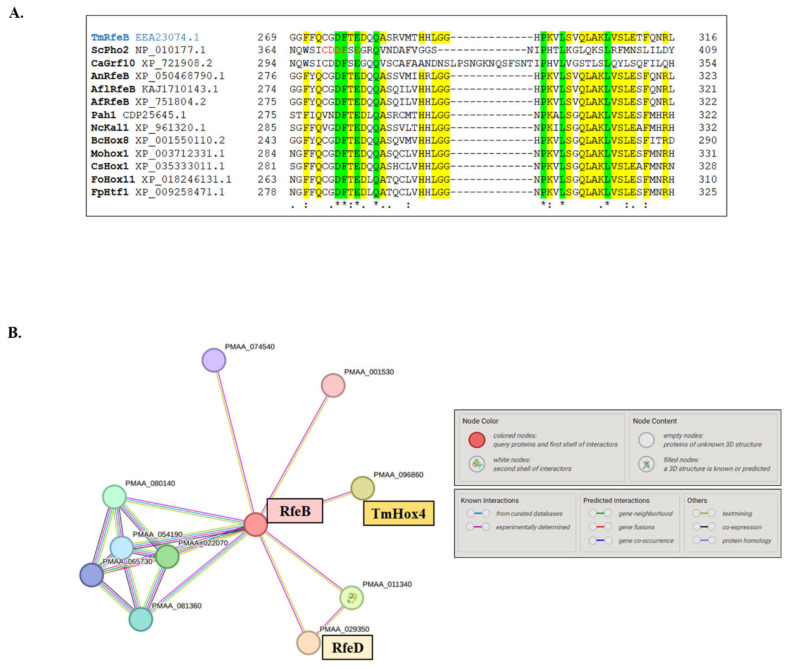
Protein-protein interaction analysis of the homeobox RfeB protein. (**A**) The homeobox RfeB protein family contained the highly conserved residues in the interaction region (IR). Protein sequences were retrieved from the NCBI database, and protein sequence alignment was performed using the Clustal W program. Red highlights the core residues of the *S. cerevisiae* Pho2 interaction region, where point mutations in these positions inhibit interactions with its three coregulators. The green highlight indicates residues conserved in all fungal species; the yellow highlight indicates residues conserved in true mold and *T. marneffei* but not in yeast *S. cerevisiae* and *C. albicans*. The blue highlight indicates the Pho2 homolog from *T. marneffei* (RfeB). Tm, *Talaromyces marneffei*; Sc, *Saccharomyces cerevisiae*; Ca, *Candida albicans*; An, *Aspergillus nidulans*; Afl, *Aspergillus flavus*; Af, *Aspergillus fumigatus*; Pa, *Podospora anserine*; Nc, *Neurospora crassa*; Bc, *Botrytis cinerea*; Mo, *Magnaporthe oryzae*; Cs, *Colletotrichum scovillei*; Fo, *Fusarium oxysporum*; Fp, *Fusarium pseudograminearum*. The symbols below the alignment indicate the conservation: an asterisk (*) indicates a single fully conserved amino acid, a colon (:) indicates amino acid groups with strongly similar properties, and a dot (·) indicates groups with weakly similar properties. (**B**) The RfeB interaction network was predicted and showed enrichment in DNA-binding transcription factor activity (GO:0003700). The amino acid sequences of RfeB were subjected to STRING analysis.

**Figure 7 jof-10-00687-f007:**
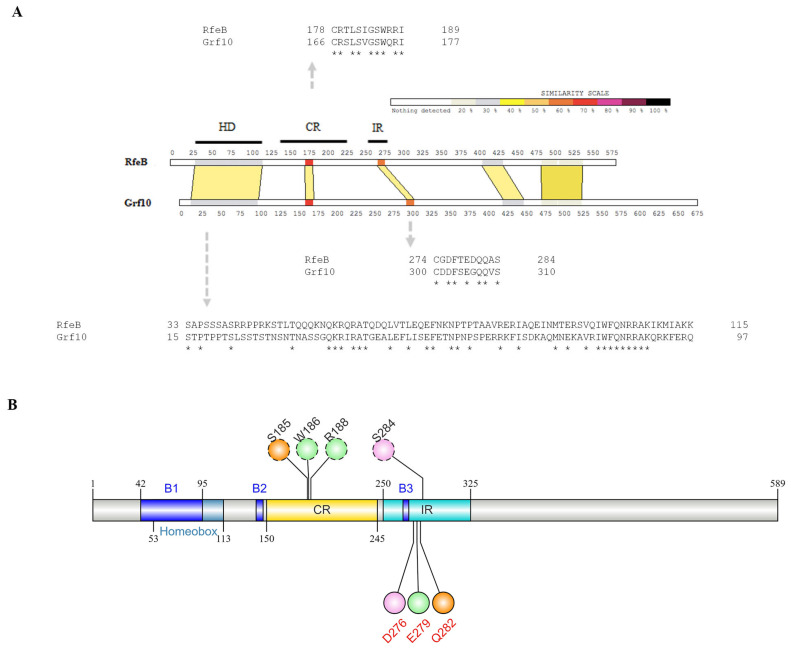
Putative functional domains of RfeB were obtained from protein alignment and molecular docking analyses. (**A**) RfeB from *T. marneffei* was aligned with Grf10 from *C. albicans*. Similarity of sequences is indicated by the scale at the top. The aligned region with identity over 35% is depicted in a tan-shaded area. The black lines indicate the regions of RfeB defined based on the corresponding functional domains from Grf10, as follows: HD, DNA-binding homeodomain; CR, central region; IR, interaction region. Conserved residues for each region (grey arrow) are shown: An asterisk (*) indicates a single fully conserved amino acid. (**B**) Schematic illustrates the conserved regions and newly identified regions/key residues of RfeB; numbers along the box indicate the amino acid position. Residues that form the internal H-bond together were depicted in the same color. Newly mapped regions are shown in blue box: B1–B3, binding site 1, binding site 2, and binding site 3, respectively. The conserved IR residues that were subjected for in silico alanine mutagenesis were highlighted in red.

**Figure 8 jof-10-00687-f008:**
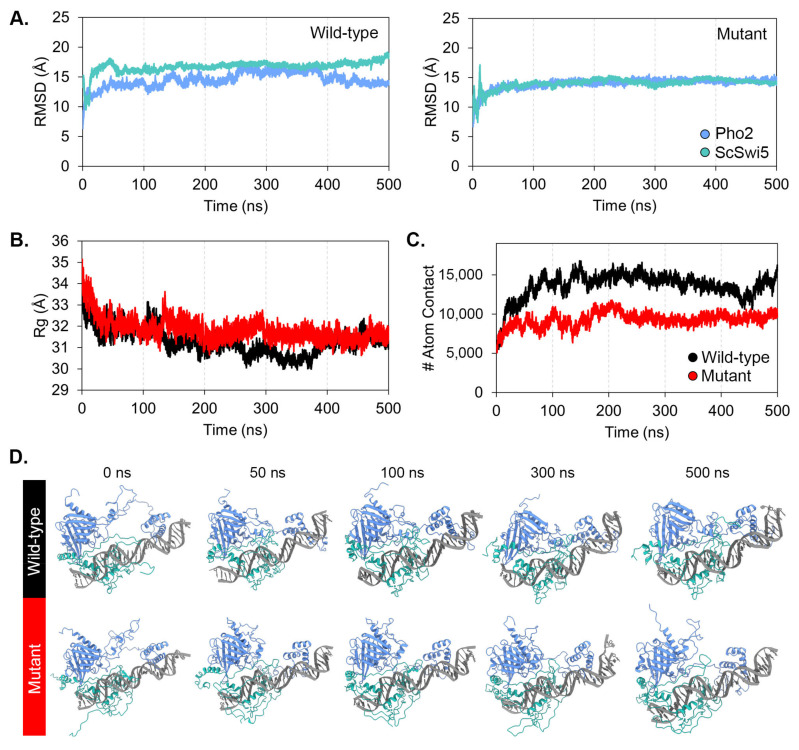
Molecular dynamics of Pho2 variants with DNA and ScSwi5. Simulation of wild-type and mutated Pho2 protein in complex with DNA and ScSwi5 were simulated for 500 ns. (**A**) Root-mean-square deviation (RMSD) of protein backbones was assessed to evaluate the stability of each protein-protein interaction. (**B**). The radius of gyration (Rg) was determined to assess the compactness of protein-protein complexes. (**C**) A number of atom contacts between Pho2 and ScSwi5 were illustrated. (**D**) The structure of the Pho2-ScSwi5 complex over time was extracted at 0, 50, 100, 300, and 500 ns.

**Figure 9 jof-10-00687-f009:**
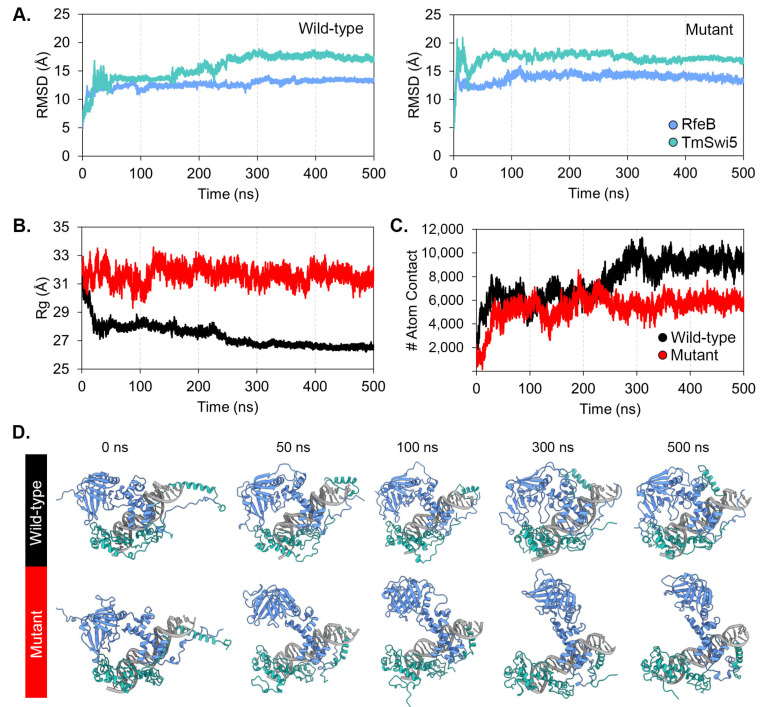
Molecular dynamics of RfeB variants with DNA and TmSwi5. Simulation of wild-type and mutated RfeB proteins in complex with DNA and Swi5 were simulated in 500 ns. (**A**) Root-mean-square deviation (RMSD) of protein backbones was assessed to evaluate the stability of each protein-protein interaction. (**B**) The radius of gyration (Rg) was determined to assess the compactness of protein-protein complexes. (**C**) A number of atom contacts between RfeB and TmSwi5 was illustrated. (**D**) The structure of the RfeB-TmSwi5 complex over time was extracted at 0, 50, 100, 300, and 500 ns.

**Figure 10 jof-10-00687-f010:**
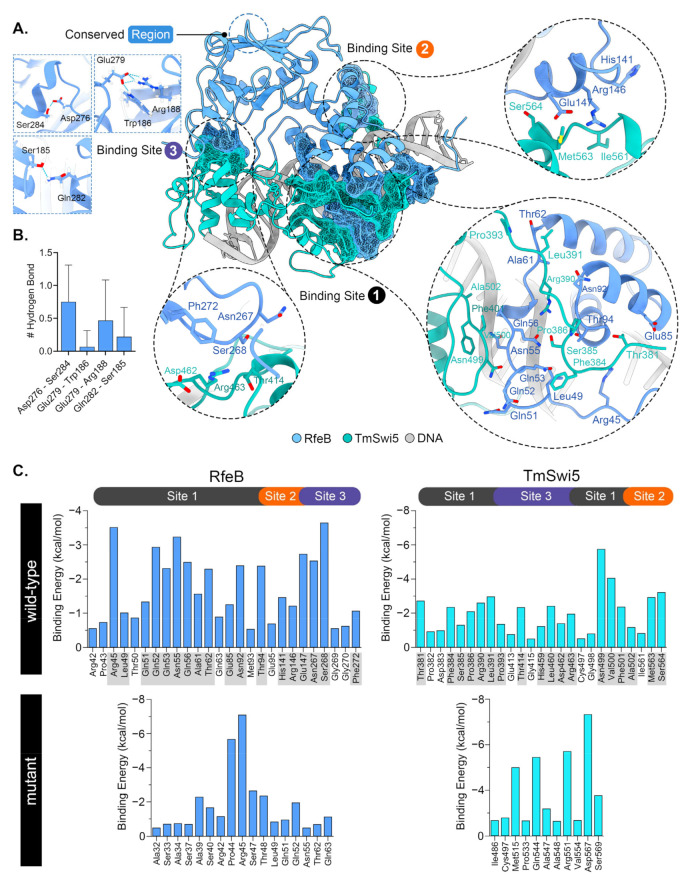
Prediction of potential residues that contribute to protein binding and stability. Residues essential for intra (**A**,**B**) and inter (**C**) protein interactions were determined. Binding sites of RfeB that are directly in contact with TmSwi5 were identified and illustrated. (**B**) Hydrogen bonds formed by the conserved IR residues (Asp276, Glu279, and Gln282) were determined. (**C**) Per-residue energy decomposition was calculated and compared between the ternary complexes formed by the wild type (top) vs. mutated RfeB proteins (bottom). Highlighted in grey were the key RfeB and TmSwi5 residues that interact with each other (binding energy less than −1 kcal/mol).

**Table 1 jof-10-00687-t001:** The roles of homeobox genes in fungal development, morphogenesis, metabolism, and pathogenicity.

Division	Species	Genes	Key Findings	Reference
Ascomycota	*A. flavus* (Mycotoxigenic mold)	*hbx1-8*	The *hbx1* gene is required for production of conidia, sclerotia, and aflatoxin.	[21,22]
	*A. fumigatus * (Human pathogen)	*hbxA*	The *hbxA gene* regulates asexual development, secondary metabolism, and virulence.	[23]
	*A. nidulans* (Mold model)	*hbxA-H*	- The *hbxA* and *hbxB* genes regulate asexual and sexual development. - The *hbxB* gene regulates trehalose biosynthesis and stress tolerance. - The *hbxD* gene is important for normal conidial production. - The *hbxA* gene regulates the expression of other key regulators involved in developmental processes, secondary metabolism, and virulence.	[24,25]
	*Botrytis cinerea* (Plant pathogen)	*BcHOX1-8*	The *BcHOX8* gene regulates morphology and pathogenicity.	[30]
	*C. albicans* (Human pathogen)	*GRF10*	- The *GRF10* gene is required for morphogenesis and pathogenicity. - The *GRF10* gene regulates the response to purine, phosphate, iron and copper.	[27,28,29]
	*Colletotrichum scovillei* (Plant pathogen)	*CsHOX1-10*	- The *CsHOX1* gene is required for host defense suppression. - The *CsHOX2* gene is important for conidiation and appressorium development.	[31]
	*Colletotrichum orbiculare* (Plant pathogen)	*CoHOX1-9*	The *CoHOX3* gene is essential for appressorium formation and pathogenicity.	[32]
	*Fusarium* spp. (Plant pathogen)	*FgHTF1* *FvHTF1* *FoHTF1*	The *Fusarium HTF1* gene is required for phialide development and conidiogenesis.	[33]
	*Magnaporthe oryzae* (Plant pathogen)	*MoHOX1-8*	The *MoHOX2* (*HTF1*) and *MoHOX7* genes are required for conidiation, appressorium development, and pathogenicity.	[34,35]
	*Neurospora crassa* (Mold model)	*kal-1*	The *kal-1* gene is necessary for basal hyphal growth, conidiation, and nutrient sensing.	[26]
	*Podospora anserina* (Mold model)	*pah1-7*	- The *pah1* gene is involved in hyphal morphogenesis and production of microconidiation. The *pah1* has no role in sexual development after fertilization. - The *pah2* and *pah5* genes play important role in shaping fruiting body and sexual asci spore formation.	[14,36]
	*S. cerevisiae * (Yeast model)	*PHO2*	- *PHO2* controls sporulation and bud morphogenesis. *PHO2* regulates the cellular response to phosphate and adenine starvation. - *PHO2* activates the gene involved in mating type switching during vegetative cell division.	[37,38,39]
	*S. pombe* (Yeast model)	*phx1*	The *phx1* gene is required for sporulation and regulates thiamine responsive genes.	[40]
	*Ustilaginoidea virens* (Plant pathogen)	*UvHOX2*	The *UvHOX2* gene regulates chlamydospore formation, conidiation, and pathogenicity.	[41]
	*Yarrowia lipolytica* (Dimorphic yeast model)	*HOY1*	The *HOY1* gene regulates filamentation.	[42]
Basidiomycota	*Volvariella volvacea* (Mushroom)	*VvHox1-8*	Gene expression analyses shown that *VvHox* genes are differentially expressed during each step of fruiting body development.	[15]
	*Ustilago maydis* (Plant pathogen)	*bE* and *bW*	- During sexual reproduction, the genes for bE/bW complex is required for stable dikaryon formation, hyphal proliferation, and pathogenicity. - In haploid cells, the genes for bE/bW complex is necessary for filamentous growth in response to nitrogen starvation.	[43,44,45]
	*C. neoformans* (Human pathogen)	*SXI2a* and *SXI1α*	- The *SXI2a* and *SXI1α* genes are required for sexual development. - Sxi2a and Sxi1α proteins regulate genes involved in sexual development, nutrient starvation, and virulence.	[20]
	*Schizophyllum commune* (Mushroom)	*hom1* and *hom2*	- The *hom1* and *hom2* genes regulate fructification process. - The *hom1* gene stimulates biomass production while the *hom2* gene represses vegetative growth.	[17,46]
	*Coprinopsis cinerea* (Mushroom)	RNA-seq analyses	Many homeobox genes show changes in expression levels during fruitification.	[16]

**Table 2 jof-10-00687-t002:** Role of *STE12* and homologous genes.

Division	Species	Genes	Key Findings	Reference
Ascomycota	*S. cerevisiae*	*STE12*	Role in mating and pseudohyphal/invasive growth.	[19]
	*C. albicans*	*CPH1*	Role in mating, filamentation, biofilm formation, and virulence.	[47,48]
	*C. glabrata*	*STE12*	Required for virulence and nitrogen starvation-induced filamentation.	[49]
	*Fusarium graminearum*	*FgSte12*	- Required for virulence and secretion of cellulase and protease. - Not involved in mycelial growth.	[50]
	*A. nidulans*	*steA*	- Required for sexual reproduction. - Not involved in mycelial growth.	[51]
	*T. marneffei*	*stlA*	- The *stlA* gene can complement the sexual defect of an *A. nidulans steA* mutant. - No role on vegetative growth, asexual development, and dimorphic switching.	[52]
	*A. oryzae*	*steA*	- Involved in regulating the expression of cell wall-degrading enzymes. - Role in cell fusion	[53,54]
	*Alternaria alternata* (Plant fungal pathogen)	*STE12*	- Role in pathogenicity. -No role in growth, ACT toxin production, and response to oxidative and osmotic stress.	[55]
	*N. crassa*	*STE12/pp-1*	Role in aerial hyphae/colony growth, conidiophore development and fertility.	[56]
	*Drechslerella dactyloides* (Nematophagus fungus)	*DdaSTE12*	Role in nematode trap formation, ring cell inflation, conidiation and vegetative growth.	[57]
	*Arthrobotrys* *oligospora* (Nematophagus fungus)	*AoSte12*	- Role in nematode trap morphogenesis, conidiation, hyphal fusion and mycelial growth. - Involved in stress tolerance response and production of secondary metabolites.	[58]
	*Setosphaeria turcica* (Plant pathogen)	*StSte12*	Role in pathogenicity by regulating vegetative growth, conidiation, appressorial development, and penetration.	[59]
	*M. oryzae*	*MST12*	- Regulating infectious growth and pathogenicity. - No role in mycelial growth.	[60]
	*B. cinerea*	*STE12*	Role in radial growth, pigment production, sclerotia formation, and pathogenicity.	[61]
	*Colletotrichum lagenariu* (Plant pathogen)	*CST1*	Role in pathogenicity by controlling the production of infectious hyphae from appressoria.	[62]
	*Cryphonectria parasitica* (Plant pathogen)	*CpST12*	Role in virulence and female fertility.	[63]
	*Metarhizium rileyi* (Entomopathogenic fungus)	*MrSte12*	- Role in virulence, appressorium formation, hyphal morphogenesis, and conidiogenesis. - Role in stress tolerance.	[64]
	*Trichoderma atroviride* (Plant fungal pathogen)	*STE12*	Role in carbon source-dependent growth, hyphal fusion, lytic enzyme expression, and mycoparasitic activity.	[65]
	*Trichoderma reesei* (Industrial fungus)	*STE12*	- Regulation of cellulase gene expression, carbon source utilization, and secondary metabolites. - Involvement in iron homeostasis via the regulation of iron transport genes and siderophore-associated genes.	[66]
	*Epichloë festucae* (Endophytic fungus)	*STE12*	Role in pathogenicity, but not hyphal fusion.	[67]
	*F. oxysporum*	*STE12*	Required for invasive growth and virulence, but not hyphal fusion.	[68,69]
	*Metarhizium acridum* (Entomopathogenic fungus)	*MaSte12*	Role in pathogenicity by regulating the appressorium formation.	[70]
Basidiomycota	*C. gattii* (Human fungal pathogen)	*Ste12alpha*	Role in mating, melanin production, and ecological fitness.	[71]
	*Puccinia striiformis f.* sp. *tritici* (Plant fungal pathogen)	*PstSTE12*	- Role in pathogenicity, haustorium formation, fungal colonization and hyphal development. - The *PstSTE12* gene can complement the mating defect of a *S. cerevisiae ste12* mutant.	[72]
	*Flammulina filiformis* (Mushroom)		- Role in fruiting body development. - Role in tolerance to salt stress, cold stress and oxidative stress.	[73]

**Table 3 jof-10-00687-t003:** List of primers used in this study.

Primer Name	Type	Sequences	Gene ID	Name
q-HOX1-F q-HOX1-R	Forward Reverse	GTCGAAAATCGAAGCCGCTC TCTCAGTGTTGTTCGACGGG	PMAA_076490	EEA26588.1 (Hox1)
q-HOX2-F q-HOX2-R	Forward Reverse	AGTCAGTGCACTTGGCTCTC AGGGGCTCTACAAAAGGCAC	PMAA_074540	EEA26382.1 (Hox2)
q-HOX3-F q-HOX3-R	Forward Reverse	CGAGCACTCCCTGACCATTT GGTACTCGTCACCAGGGTTG	PMAA_085630	EEA24568.1 (Hox3)
q-HOX4-F q-HOX4-R	Forward Reverse	ACCCAGCATCTTCGCCTATG GCACGCAAGACGTCTGTAAC	PMAA_096860	EEA23091.1 (Hox4)
qRfeB F qRfeB R	Forward Reverse	CCAGATTCAATGCGGCAATAC TACTTAGTCCACCGAGTCCAT	PMAA_096690	EEA23074.1 (Hox5)
q-HOX6-F q-HOX6-R	Forward Reverse	ACAGCTCATCGCTATCACGG GGACGGGGATGTTTGTCTGT	PMAA_043800	EEA20547.1 (Hox6)
q-HOX7-F q-HOX7-R	Forward Reverse	TCGTCCCTGGACTCTGACAT TTGGGGACGTTGTTGCTTCT	PMAA_001530	EEA19358.1 (Hox7)
act1-F act1-R	Forward Reverse	TGATGAGGCACAGTCTAAGC CTTCTCTCTGTTGGACTTGG	PMAA_012310	EEA18960.1 (Actin)

**Table 4 jof-10-00687-t004:** List of putative transcription factors that contain the homeobox domain IPR001356 in *T. marneffei*.

Designated Hox Name	Name	Protein ID
TmHox1	YOX1	EEA26588.1
TmHox2	-	EEA26382.1
TmHox3	-	EEA24568.1
TmHox4	Homeobox protein TGIF2LX	EEA23091.1
TmHox5	RfeB/Hoy1	EEA23074.1
TmHox6	-	EEA20547.1
TmHox7	Homeobox protein AKR	EEA19358.1

**Table 5 jof-10-00687-t005:** Expression pattern of *T. marneffei* homeobox genes that were previously examined by transcriptomic studies.

Protein	Strain ID		Alternative Name	^C^ Pasricha et al., 2013 [96]						^C^ Yang et al., 2014 [95]	^C^ Lin et al., 2012 [97]			
	ATCC 18224 ID	PM1 ID		Fold Change (log_2_)			% of Total Expression			^b^ Pattern	Fold Change 37 vs. 25	Pattern 37 vs. 25	37-log2	25-log2
				^a^ 37 > 25 vs. 37	25 > 37 vs. 25	37 vs. 25	Hyphal	Asexual	Yeast					
EEA26588.1 (Hox1)	PMAA_076490	GQ26_0112560	Yox1	na	na	na	na	na	na	0001	2.57	up	16.53	15.17
EEA26382.1 (Hox2)	PMAA_074540	GQ26_0110650		na	na	na	na	na	na	0100	na	na	na	na
EEA24568.1 (Hox3)	PMAA_085630	GQ26_0280750		na	na	na	na	na	na	1100	25.87	down	7.48	12.17
EEA23091.1 (Hox4)	PMAA_096860	GQ26_0160760	TGIF2LX	2.2	−0.42	−0.58	17	71	12	0011	8.49	up	8.05	4.96
EEA23074.1 (Hox5)	PMAA_096690	GQ26_0161010	RfeB/Hoy1	na	na	na	na	na	na	1011	na	na	na	na
EEA20547.1 (Hox6)	PMAA_043800	GQ26_0142040		na	na	na	na	na	na	0011	6.98	down	10.64	13.44
EEA19358.1 (Hox7)	PMAA_001530	GQ26_0024010		na	na	na	na	na	na	0111	7.85	down	13.42	16.40
EEA26628.1	PMAA_015520			na	na	na	na	na	na		93.57	up	10.3	3.75
EEA28535.1	PMAA_033420			na	na	na	na	na	na		na	na	na	na
		GQ26_0480540	Homeobox protein Meis3-like 1	na	na	na	na	na	na	0100	na	na	na	na
EEA24489.1 (SteA)	PMAA_084900	GQ26_0530470	Ste12, StlA	na	na	na	na	na	na	1111	na	na	na	na

^a^ = Indicate temperature (37 °C or 25 °C) for culture growth. ^b^ = A four-digit code denotes the expression pattern of the gene from four experimental treatments: (1) stable growth at 37 °C as yeasts (stable yeast, Y), (2) yeasts grown at 37 °C transferred to 25 °C for 6 h (yeast-to mycelium, Y-to-M), (3) mycelia grown at 25 °C transferred to 37 °C for 6 h (mycelium-to-yeast, M-to-Y), and (4) stable growth at 25 °C as mycelia (stable mycelium, M). ‘‘1’’ indicates high expression or ‘‘0’’ indicates low expression of a gene under the four treatments. Note the expression levels of a gene were compared between treatments of the same gene, not against expression levels of other genes (Yang et al., 2014 [95]). na = Gene was not examined. ^C^ Pasricha et al., 2013 = Strain: ATCC18224, Stable form: 25 °C on ANM plates for asexual development (4 d), agitated in liquid BHI for hyphal growth at 25 °C (2 d), and yeast growth at 37 °C (4 d of growth followed by 10 mL transferred to fresh medium for an additional 2 d), Phase transition: 6-h. ^C^ Yang et al., 2014 = Strain: PM1, Stable form: 7 days at 25 °C or 37 °C on SDA plates, Phase transition: 6-h. ^C^ Lin et al., 2012 = Strain: B-6323 isolated from a Chinese patient with talaromycosis skin lesions, Stable form: Sabouraud glucose broth at 25 °C or 37 °C for 48-h.

**Table 6 jof-10-00687-t006:** Relative binding free energy between RfeB-TmSwi5 and Pho2-ScSwi5 from MM-GBSA calculations.

Component	Free Energy (kcal/mol)			
	RfeB-TmSwi5		Pho2-ScSwi5	
	Wild-Type	Triple Mutant	Wild-Type	Triple Mutant
Gas phase				
Van der Waals	−177.96 ± 10.13	−104.77 ± 7.05	−241.17 ± 17.45	−175.65 ± 9.08
Electrostatic	1387.77 ± 100.32	1960.49 ± 121.79	−1262.06 ± 138.54	−124.02 ± 110.26
Solvent phase				
Polar solvation	−1281.85 ± 94.0	−1875.47 ± 116.85	1437.11 ± 134.07	259.44 ± 106.89
Non-polar solvation	−24.94 ± 1.12	−15.47 ± 0.99	−35.05 ± 2.46	−25.51 ± 1.22
Total	−96.98 ± 11.86	−35.22 ± 9.55	−101.17 ± 22.05	−65.74 ± 11.2

## Data Availability

Gene expression datasets generated and analyzed during the current study are deposited and available at Figshare (DOI: 10.6084/m9.figshare.25241935). Data used to generate a heatmap in Figure 5 are also freely accessible at Figshare (DOI: 10.6084/m9.figshare.25241935). Primer sequences are provided in Table 3.

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
