# Peer review of "Identification of Homeobox Transcription Factors in a Dimorphic Fungus Talaromyces marneffei and Protein-Protein Interaction Prediction of RfeB"

_jof, 2024, doi:10.3390/jof10100687_

Round 1

Reviewer 1 Report

Pongpom et al. address the question of the potential roles of homeobox transcription factors in the biology of Talaromyces marneffei, a dimorphic ascomycete that causes mycoses. The work is introduced well and is overall well written. The structure figures are clear. Nevertheless, I had multiple concerns about the scientific approaches taken that seriously reduced my confidence in the results obtained and the interpretations of their significance. These concerns could be remedied by incorporating controls into the studies and being more circumspect in interpretation.

Identifying homeobox proteins through the Fungal Transcription Factor Database and subsequent bioinformatic analyses and comparisons is potentially important to consider but not by itself particularly significant. Six of the seven genes that are focused on in this study are already annotated as Hox genes in FungiDB, and the seventh has obvious Hox domains in that database. It might be appropriate for the authors to compare their results for the larger HOX set with the FungiDB annotations. All have developmental transcriptomics data associated with them that are also included in that database.

What could bring significance to these findings are analyses that provide context for how the putative proteins function in the organism. Here, RT-qPCR is used to examine relative gene expression levels of six bioinformatically identified transcripts under different developmental conditions to begin to address this question. There are no controls in this analysis – no analyses or consideration of genes previously identified as differentially expressed in yeast, mycelia, or conidia to use as references for expression. Only actin is used as a control (note, however, it is missing in Table 2), and the basis for it having constant expression is not given, so how it serves as a control is unclear. The published global RNA-seq analyses of T. marneffei transcripts (PMC4199489) during development is not considered or referenced. But, since the transcripts examined here were also examined in that previous study, shouldn’t it be mentioned how they behaved in that study, and how do some signature transcripts described there behave in these studies? I also note a difference in that previous study for how conidia, yeast and mycelial phases are obtained relative to this study – are there ways to make direct comparisons for transcript induction conditions with any previous studies? In conclusion, I cannot evaluate the reported results with any confidence for their significance without comparisons to controls since this is not a transcriptome-wide study (and those studies take retrospective looks in any case) – even images of cells at the different developmental stages are not included. Also, given the limited data presented (a single northern blot) on TmStlA in an early study, why was it deliberately excluded on the basis that it was already studied, since this is clearly a critical Hox gene in fungal development?

With respect to the predicted structure data (AlphaFold3) and molecular modeling data, which is based on Talaromyces protein predictions based on Supp Data S2 and by inference (the text should call the proteins TmRfeB and TmSwi5 and not make a disclaimer not to do so, since many different sources of these proteins are discussed in the structure section of the manuscript). First: no data directly support the interaction of TmRfeB and TmSwi5 as I understand it (e.g., two-hybrid data). With the disclaimer that I am not an expert in assessing molecular dynamics simulations, I will accept the results as stated. But with that said, there are no controls here in the sense that there is extensive literature on known Hox interactions with other proteins, but these interactions, and known effects of mutations on factor function, are not tested using the same parameters used to look at the Tm proteins. Therefore, even treating the tools as “black boxes,” I have reduced confidence in the value of the reported results since the “black boxes” as used are not established to give data consistent with corroboratory evidence obtained by other means.

Specific comments

P2

Kal-1 here is a protein, don’t italicize

The Gfr10 -> Gfr10

P3-4 “The” not needed in front of proteins. Gene and protein conventions are intermixed here, be explicit – all genes, or all proteins

P5

L113 Presumed -> this is what is done, but it is not appropriate without additional data

L126 count -> determine

L130 and elsewhere: use xg, not rpm

L6

L177 modeled/modeling: rewrite

L182-183: is this particularly complicated or were subsequent analyses? Why weren’t individual Ala substitutions analyzed, or other amino acid substitutions?

L8

L240 rationale for excluding?

P9

L259 says only values greater than 40% are shown… should it say values greater than 40% are shown with a green dot…

L265 I did not understand why StlA was excluded. Also, include its EEAxxxxx designation

P12

L322-324 I don’t see the basis for this assertion. The results suggest that they are differentially expressed.

L359 Tm has a putative PalcA homolog PMAA_051940 which is considered to be a Pho4 ortholog in other fungi, correct?

Section 3.6

I found the discussion of the structures as if they were real disconcerting, especially in the context of the concern raised above, which is there is no experimental evidence to support this modeling works by comparing it to a system where there is a real structure :

For examples, “Additionally, the ternary complex stability was evaluated”; “the number of atomic contacts between these proteins was monitored”

P21

L477 Disconcerting: Determination of key residues that contribute to protein binding and stability. This is a hypothetical model. That is not bad. But it is not validated like a cryoEM structure or even biochemical interaction studies. And in any case, other residues were not altered, correct, nor were these changed singly: so do we know all of these are key residues, even in the model?

L511: undrugable-> undruggable

L522 can be found in other excellent review -> has been reviewed

L525 Sisyphean task – the rock always rolls down – I don’t think this is the allegorical point – possibly Herculean task?

Author Response

Please also the attachment file name: Response to reviewer 1 JoF 11Sep24 for the latest response.

Comment 1: It might be appropriate for the authors to compare their results for the larger HOX set with the FungiDB annotations. All have developmental transcriptomics data associated with them that are also included in that database.

Response 1: Data associated with identified HOX genes from genetic studies and gene expression analyses from other fungal species have been collected in Table 1 and Table 2 (revised manuscript, also see comment in response # 3). We believe that data in Table 1 and Table 2 represent one of the most recent and comprehensive literature reviews of experimentally verified fungal homeobox genes. In addition, we have compared our data to the homeobox gene studies from straw mushroom V. volvacea, and plant pathogens Fusarium spp as appeared in discussion.

Comment 2: There are no controls in this analysis – no analyses or consideration of genes previously identified as differentially expressed in yeast, mycelia, or conidia to use as references for expression. Only actin is used as a control (note, however, it is missing in Table 2), and the basis for it having constant expression is not given, so how it serves as a control is unclear.

Response 2: First, the normalized expression levels of a homeobox gene were compared between cell types (conidia, yeast, mold) or phase transition (conidia transition to yeast or conidia transition to mold) of the same gene, not against expression levels of other genes. This is to examine if the putative homeobox genes are expressed or not in different developmental stages of T. marneffei. We stated in Figure 4 legend that, “relative fold-change levels were compared to the phase where the transcript was the lowest”. To make this sentence clearer, we edited the above sentence to the following sentence.

Fold-change levels were normalized to 1 for the phase where the transcript was the lowest.

Also, we added following sentence to Figure 5 legend

Fold-change levels of conidial cells were normalized to 1. 

Second, our laboratory has previously determined four housekeeping genes that be potentially used as endogenous control for expression analysis by qRT-PCR. These genes are as follows:  β-actin (act); glyceraldehyde-3-phosphate dehydrogenase (gapdh); β-tubulin (benA) and 18S rRNA. This study indicates that actin gene is the most suitable reference gene to use in T. marneffei qRT-PCR experiment across different cell types or stress conditions (Dankai et al. 2015). We added this reference and primer sequences for actin gene in Table 3 (revised manuscript).

Third, we did not examine genes previously identified as differentially expressed in yeast, mycelia, or conidia to use as references for expression. However, the actin gene itself always served as the gene expression control since it is differentially expressed among the mold, conidia and yeast phase of this dimorphic fungus as shown in Kummasook et al. study (reference 77 in revised manuscript). We obtained cultures in conidia, mold phase and yeast phase following our previously published protocol and routinely culture protocol.

Comment 3: The published global RNA-seq analyses of T. marneffei transcripts (PMC4199489) during development is not considered or referenced. But, since the transcripts examined here were also examined in that previous study, shouldn’t it be mentioned how they behaved in that study, and how do some signature transcripts described there behave in these studies? I also note a difference in that previous study for how conidia, yeast and mycelial phases are obtained relative to this study – are there ways to make direct comparisons for transcript induction conditions with any previous studies?

Response 3: For comparison purposes, we added a new Table 5 (revised manuscript) to describe expression patterns of homeobox genes from previous transcriptomic studies in T. marneffei. As you noted, different T. marneffei strains and culture growth conditions were used for each study (Table 5). Our study used T. marneffei strain with Thailand origin (ATCC200051); Yang et al. 2014 used T. marneffei PM1 strain; Pasricha et al. 2013 used T. marneffei ATCC18224 strain; and Lin et al. 2012 used T. marneffei B-6323 strain. Despite a variation in experimental parameters, comparative analyses (Table 5, Figure 4 and Figure 5) suggest that the gene expression patterns of most homeobox genes are consistent with at least one of the previously published data. The manuscript was edited as shown below (highlighted in red) to describe how homeobox genes behaved in previous studies compared to our current study (please see result 3.3, page 13-15 in the revised manuscript).

            To investigate the biological relevance of putative TmHox genes in vitro, we examined the transcript levels of TmHox genes from T. marneffei ATCC200051 strain growing in three different conditions: on PDA plates for conidiation (asexual development), in liquid SDB for hyphal growth at 25°C, and yeast growth at 37°C. Gene expression patterns of seven TmHox genes were classified into three different groups. Group I was comprised of the conidia specific gene TmHox4 (Figure 4A). The TmHox4 transcripts were highly accumulated during conidiation, being 500-fold higher than transcript levels presented in the hyphal or yeast phases. This result is consistent with DNA microarray study by Pasricha et al. 2013, showing that TmHox4 gene homolog from T. marneffei ATCC18224 strain was accumulated at high levels in asexual conidial cell (Table 5; Pasricha et al. 2013). Group II was classified as genes whose expression levels were strongly downregulated in conidia, but upregulated in mold and/or yeast forms, including TmHox1, TmHox5 (RfeB), and TmHox7 (Figure 4B). Upregulated levels of TmHox1 gene in stable mycelial form relative to stable yeast form have been detected in previous transcriptomic study (Table 5; TmHox1 (Yang et al. 2014)). Also, the RfeB gene is highly expressed in both stable mycelial and yeast form, which is in agreement with RNA-sequencing data by Yang et al. 2014 (Yang et al. 2014). Lastly, group III consisted of genes that were expressed at low levels in the yeast phase and were differentially upregulated in the mycelium phase, including TmHox2, TmHox3 and TmHox6 (Figure 4C). This expression pattern of TmHox3 and TmHox6 genes has been previously reported from different T. marneffei strains (TmHox3 (Lin et al. 2012); TmHox6 (Yang et al. 2014 and Lin et al. 2012)).

(Page 15, line 349) TmHox4, TmHox2, and RfeB genes were greatly upregulated during the mycelium phase transition. For instance, TmHox2 gene was significantly upregulated 7-fold when transitioning from conidia to mycelium phase (25°C, 2-hr) (Figure 5B). The study by Yang et al. 2014 found that TmHox2 gene was highly expressed during yeast to mycelium phase switching (Table 5; Yang et al. 2013). Additionally, our result on TmHox4 gene is consistent with data by Pasricha et al. 2013, demonstrating that TmHox4 gene was upregulated during phase transition from yeast (37°C) to hyphal (25°C) phases (Table 5; [96]). In contrast, TmHox6 transcript was highly accumulated in conidia and its transcript levels were decreased during the early yeast and mycelium phase transitions (Figure 5C). These transcript profiles suggested that TmHox4, TmHox2 and RfeB could be involved in hyphal growth during the early stage. 

Comment 4: In conclusion, I cannot evaluate the reported results with any confidence for their significance without comparisons to controls since this is not a transcriptome-wide study (and those studies take retrospective looks in any case) – even images of cells at the different developmental stages are not included.

Response 4: This point has been resolved in our revised manuscript. On the point about the expression control, the actin gene itself already served as the differential control since this gene is normally expressed differently in each phase as shown in our previous study (Kummasook et al. 2010 [1]). In short, we used Northern blot analysis to show that this gene has different basal level of transcripts in conidia (dormant phase), mycelia and yeast (metabolic active phases). The growth conditions for generating conidia, stable mold phase and stable yeast phase of T. marneffei strain ATCC200051 have been previously optimized and established by our laboratory (Kummasook et al. 2010 [1]). Attached here is the image of cells at different developmental stages from Kummasook et al. 2010 We decided not to include the image of cells at different developmental stages, but provide the reference in the revised manuscript, since it was already been published.

Therefore, the following modified sentence was added into the main manuscript under materials and methods.

Preparation of T. marneffei in different growth forms was conducted as previously described (Kummasook et al. 2010). Briefly, the conidia were harvested and 1x108 conidia/ml were inoculated in a 50-ml Sabouraud’s dextrose broth (SDB). Cultures were incubated at 25°C (mycelium phase) and 37°C (yeast phase) with continu-ous shaking at 200 rpm.  After 72 hours, the stable mycelium and yeast cultures were collected by centrifugation at 4°C, 7,000 rpm for 30 min.

Comment 5: Also, given the limited data presented (a single northern blot) on TmStlA in an early study, why was it deliberately excluded on the basis that it was already studied, since this is clearly a critical Hox gene in fungal development?

Response 5: Our original aim was to test the expression of uncharacterized homeobox genes to gain insights into the biological relevance of bioinformatically identified genes. On the other hand, the TmStlA gene has been identified and characterized its functions by genetic study and expression analysis (Borneman et al. 2000 [2]). To emphasize the critical functions of STE12 proteins and complete the full review of homeodomain-containing transcription factor in fungi, we generated Table 2 to summarize data from experimentally verified STE12 genes. The following modified sentence was added.

(Page 2, line 70-72, revised manuscript) Strikingly, this homeodomain-containing transcription factor family plays a significant role in the regulation of sexual reproduction, morphogenesis, and metabolism across fungal members of Ascomycota and Basidiomycota (Table 1 and Table 2).

Also, we are able to compare the TmStlA gene expression data by Borneman et al. 2000 with transcriptomic data by Yang et al. 2014. The following sentence was added into the main text.

(Page 12, line 281, revised manuscript) For further analysis, we focused on seven uncharacterized genes that contain the homeobox domain IPR001356, designated as TmHox1 – TmHox7 (Table 3, Figure 3). We excluded the homeobox EEA26628.1 and EEA28535.1 proteins because they contain only the conserved site of the homeobox protein, antennapedia type (IPR001827) while the homeobox protein domain IPR001356 is absent (Figure 3, Supplemental Data S1). Also, the StlA gene was previously characterized in T. marneffei to be expressed primarily in vegetative tissues, and therefore excluded from our current study [37] (Table 5; Yang et al. 2014).

Comment 6:

6.1 With respect to the predicted structure data (AlphaFold3) and molecular modeling data, which is based on Talaromyces protein predictions based on Supp Data S2 and by inference (the text should call the proteins TmRfeB and TmSwi5 and not make a disclaimer not to do so, since many different sources of these proteins are discussed in the structure section of the manuscript).

Response 6.1: All Swi5 proteins have been renamed to either ScSwi5 or TmSwi5 throughout the revised manuscript.

6.2 First: no data directly support the interaction of TmRfeB and TmSwi5 as I understand it (e.g., two-hybrid data). With the disclaimer that I am not an expert in assessing molecular dynamics simulations, I will accept the results as stated.

Response 6.2: The main purpose of performing in silico mutagenesis and molecular dynamics simulation was to assess the importance of conserved IR residues. The only known function of IR residues comes from the study in S. cerevisiae Pho2 protein, which is to mediate protein-protein interaction. However, we did not know the protein partners of RfeB in T. marneffei. To set up the system where we could test the function of IR residues in protein interaction, we chose TmSwi5 as an RfeB model protein partner for the simulation test (see Page 4, line 479, revised manuscript for rationale of choosing TmSwi5). The molecular dynamics simulation revealed that alanine substitutions at IR residues of RfeB led to alteration in 3D structure and decrease in protein stability. We understand that our study could not conclude that TmSwi5 is the protein partner of RfeB until we performed experiments, yet we gained confidence that the RfeB IR residues have critical functions in T. marneffei.

6.3 But with that said, there are no controls here in the sense that there is extensive literature on known Hox interactions with other proteins, but these interactions, and known effects of mutations on factor function, are not tested using the same parameters used to look at the Tm proteins. Therefore, even treating the tools as “black boxes,” I have reduced confidence in the value of the reported results since the “black boxes” as used are not established to give data consistent with corroboratory evidence obtained by other means.

Response 6.3: The interaction of Pho2-ScSwi5-DNA is known and well characterized in S. cerevisiae. There is extensive literature on Pho2 interactions with ScSwi5 and known effects of mutations on Pho2 functions. Thus, we tested this Pho2-ScSwi5-DNA complex formation using the same parameters used to look at the Tm proteins (AlphaFold3 model, in silico mutagenesis and molecular dynamics). Results are shown in Figure 8, Figure S1, Figure S2, Figure S5 and Table 6 (revised manuscript).

Our results from AlphaFold3 model and molecular dynamics simulation are consistent with previously published data on Pho2 as following.

- (Page 21, Line 458 – 462, revised manuscript) The AlphaFold3 model predicted that Pho2 interacted with DNA sequence of 5´-CAATTTA-3´ and ScSwi5 interacted with sequence of 5´-AAACCAGCAT-3´ of HO promoter (Figure S1), corresponding to the previous results from methylation and hydroxy radical interference [99].

- (Page 21, Line 471 – 478, revised manuscript) As expected, the alanine substitutions at the conserved residues Asp371, Glu374, and Gln377 resulted in a decrease in the relative binding free energy and atomic contacts with ScSwi5 (Table 6 and Figure 8C), highlighting their contribution to protein-protein interactions. Overall, this result suggested that the conserved IR residues contributed to Pho2-ScSwi5 protein interaction as previously verified in S. cerevisiae [37]. Thus, AlphaFold3 and simulation systems could accurately predict protein-protein complex structure with high confidence.

One new potential function of the IR residues we gained from molecular dynamics simulation is that, the IR residues contribute to protein stability by forming internal hydrogen bonds with residues located in the central region. We found this feature in both Pho2 and RfeB complexes. Following result was obtained from Pho2-ScSwi5-DNA complex simulation.

- (Page 29, Line 659 – 666, revised manuscript) Consistently, the Pho2 residue Asp371 was also found to stabilize the α-helix and β-sheet structures by forming intramolecular hydrogen bond with Asn379 with high occupancy (Figure S5). Also, the Pho2 residue Gln377 interacted with Ser206 located in central region to stabilize Pho2 protein tertiary structure (Figure S5). Thus, the internal formation of hydrogen bonds by the IR conserved aspartate and glutamine residues were observed in both RfeB and Pho2 protein complexes.

P2

Kal-1 here is a protein, don’t italicize

The Gfr10 -> Gfr10

Response: We updated the table to use gene conventions in the revised manuscript.

P3-4 “The” not needed in front of proteins. Gene and protein conventions are intermixed here, be explicit – all genes, or all proteins

Response: Thank you. We updated the table to use gene conventions in the revised manuscript.

P5

L113 Presumed -> this is what is done, but it is not appropriate without additional data

Response: We edited the sentence by changing the word presumed to “assessed”.

L126 count -> determine

Response: We edited the sentence as suggested.

L130 and elsewhere: use xg, not rpm

Response: We edited this information as suggested.

L6

L177 modeled/modeling: rewrite

Response: We edited the sentence to the following.

The protein-protein complex was modeled, using AlphaFold3 tool.

L182-183: is this particularly complicated or were subsequent analyses? Why weren’t individual Ala substitutions analyzed, or other amino acid substitutions?

Response: We did not do individual alanine substitution analysis because we wanted to know the impact of the changes in highly conserved IR residues on overall protein structure and interaction. Indeed, in silico mutagenesis on multiple residues of protein of interest is commonly performed, such as in this study [3]. We chose alanine because it is an amino acid that is widely used for identifying protein positions that are important for function or ligand binding.

L8

L240 rationale for excluding?

Response: The reason that we excluded the homeodomain-like proteins is because we found more than 100 proteins for this type of protein. There would be too much to construct phylogenetic tree or perform gene expression analysis.

Therefore, we revised the previous sentence to the following.

We examined ten homeobox proteins in T. marneffei genome that contain homeodomain.

P9

L259 says only values greater than 40% are shown… should it say values greater than 40% are shown with a green dot…

Response: We edited the sentence as suggested.

L265 I did not understand why StlA was excluded. Also, include its EEAxxxxx designation

Response: We added EEA24489.1 for StlA protein on P. 11 Line 257.

We also changed the content in previous Line 265, please see comment #5.

P12

L322-324 I don’t see the basis for this assertion. The results suggest that they are differentially expressed.

Response: We edited the sentence as suggested.

L359 Tm has a putative PalcA homolog PMAA_051940 which is considered to be a Pho4 ortholog in other fungi, correct?

Response: We identified homologs of Bas1, Pho4 and Swi5 by using protein sequences of Pho4 from S. cerevisiae and performed BLASTP search against T. marneffei ATCC18224 strain. The conclusion for no homolog of Pho4 in T. marneffei was based on the result, stating that “No significant similarity found”. Below is the result obtained from BLAST search.

As you suggested, we searched PalcA gene in database such as fungiDB, yeastgenome, and candidagenome and found that PalcA is the Pho4 homolog in T. marneffei. We directly aligned protein sequences of Pho4 from S. cerevisiae and PalcA from T. marneffei, using blasp suite-2sequences as shown below.

Accordingly, following paragraphs were edited for better accuracy.

            (Page 2, line 437, revised manuscript) Since the CDDF-E residues are well conserved in RfeB, we hypothesized that RfeB could interact with other proteins to regulate specific downstream target genes. We performed BLAST homology search against known Pho2 protein partners from S. cerevisiae and found that T. marneffei showed the highest percentage identity with Swi5 (TmSwi5/PMAA_023500/EEA27475.1: 54.55%) while exhibited less conservation with Bas1 (Eta2/PMAA_062970EEA25176.1: 33.54%) and Pho4 (PalcA/PMAA_051940/ EEA21384.1: 31.48%). To further predict RfeB interaction network, the amino acid sequences of RfeB were subjected for STRING analysis (Figure 6B). The RfeD (PMAA_029350) and TmHox4 (PMAA_096860) transcription factors showed the highest predicted score for being a protein partner of RfeB (Figure 6B).

            (Page 4, line 479, revised manuscript) To evaluate how the conserved IR region contributes to protein complex formation, a protein-protein complex was modeled, in combination with molecular dynamics simulation. In initial analysis, TmSwi5 was selected as RfeB protein partner because TmSwi5 showed the highest homology with ScSwi5.

Section 3.6

I found the discussion of the structures as if they were real disconcerting, especially in the context of the concern raised above, which is there is no experimental evidence to support this modeling works by comparing it to a system where there is a real structure:

For examples, “Additionally, the ternary complex stability was evaluated”; “the number of atomic contacts between these proteins was monitored”

Response: The RfeB-TmSwi5 model has been compared to a system of Pho2-ScSwi5 model where the interaction is known. See response to comment #6.

P21

L477 Disconcerting: Determination of key residues that contribute to protein binding and stability. This is a hypothetical model. That is not bad. But it is not validated like a cryoEM structure or even biochemical interaction studies.

Response: Based on the computational results, we could potentially determine the important residues contributing to protein-protein interaction, as shown in Figure 10C. We agreed that our simulation result is a hypothetical model, which is not validated like a cryoEM or biochemical interaction studies. Yet, these identified potential residues, such as Ser268 that showed highest percentage of hydrogen bond occupancy with TmSwi5 and Phe272 that showed high sequence conservation in many fungal species (Figure 6A), will be important candidate for mutagenesis studies in the future.

Accordingly, we changed the Figure 10 legend to:

(P. Line 626, revised manuscript) Prediction of potential residues that contribute to protein binding and stability.

And in any case, other residues were not altered, correct, nor were these changed singly: so do we know all of these are key residues, even in the model?

Response: We did not alter other residues besides the three IR conserved residues. Also, we did not singly change one residue, but all three IR residues were replaced by alanine simultaneously. We agree with the reviewer’s opinion about the single change of any residues can be important to evaluate the specific role of each residue. Nonetheless, in silico mutagenesis on multiple residues of interest is commonly performed, such as in this study [3]. Here, our residues of interest are the conserved IR residues, and data from our model supported the critical function of IR residues (see additional response below).

For this specific point: so do we know all of these are key residues, even in the model?

Response:

            We know from the model that the three IR residues of RfeB and Pho2 contribute to proper protein folding that potentially mediates protein interaction with Swi5 proteins.

            Based on the molecular dynamics simulation of wild type proteins, we suggested that the IR conserved aspartate residue, i.e. Asp276 in RfeB and Asp371 in Pho2, is the most important residue among the three selected residues. Our model demonstrated that this specific aspartate residue stabilized the secondary structure of α-helix and the loop region by forming hydrogen bond with the next eight- residue, i.e. Asp276 with Ser284 for RfeB and Asp371 with Asn379 for Pho2 (Figure 10A and Figure S5A). Additionally, these aspartate-dependent interactions showed the highest percentage of hydrogen bond occupancy in comparison to the other two conserved residues (Figure 10B and Figure S5B). These specific interactions were absent in both mutated RfeB and Pho2 proteins, leading to the changes in secondary structure of RfeB and Pho2 proteins, and reductions in protein-protein interactions.

            We also know from the model that the IR residues did not make direct contact with Swi5 proteins. However, it is still possible that the IR residues can interact with other protein partners that we have not tested in this current study.

According to the response replying to this comment, we revised our discussion as shown below.

(Page 31, Line 750 - 754, revised manuscript)

The RfeB residues that potentially mediate protein-protein interaction include Ser268, which showed highest percentage of hydrogen bond occupancy with TmSwi5 (Figure 10C), and Phe272, which showed high sequence conservation in many fungal species (Figure 6A and Figure 10C). These potential identified residues will be important candidates for mutagenesis studies in the future.

(Page 31, Line 758 - 766)

Molecular dynamics simulation of wild type proteins suggested that the IR conserved as-partate residue, i.e. Asp276 in RfeB and Asp371 in Pho2, is the most important residue among the three selected residues. Our model demonstrated that this specific aspartate residue stabilized the secondary structure of α-helix and the loop region by forming hy-drogen bond with the next eight- residue, i.e. Asp276 with Ser284 for RfeB and Asp371 with Asn379 for Pho2 (Figure 10A and Figure S5A). Additionally, these aspar-tate-dependent interactions showed the highest percentage of hydrogen bond occupancy in comparison to the other two conserved residues for both RfeB and Pho2 models (Figure 10B and Figure S5B).

(Page 31, Line 777 - 778)

Alternatively, the CDDF-E residues can still directly interact with other protein partners that we have not tested in this current study.

L511: undrugable-> undruggable

 Response: We edited as suggested.

L522 can be found in other excellent review -> has been reviewed

Response: We edited as suggested.

L525 Sisyphean task – the rock always rolls down – I don’t think this is the allegorical point – possibly Herculean task?

Response: We edited as suggested.

Reference

  1. Kummasook, A.; Tzarphmaag, A.; Thirach, S.; Pongpom, M.; Cooper, C.R., Jr.; Vanittanakom, N. Penicillium marneffei actin expression during phase transition, oxidative stress, and macrophage infection. Mol Biol Rep 2011, 38, 2813-2819, doi:10.1007/s11033-010-0427-1.
  2. Borneman, A.R.; Hynes, M.J.; Andrianopoulos, A. An STE12 homolog from the asexual, dimorphic fungus Penicillium marneffei complements the defect in sexual development of an Aspergillus nidulans steA mutant. Genetics 2001, 157, 1003-1014, doi:10.1093/genetics/157.3.1003.
  3. Casalino, L.; Gaieb, Z.; Goldsmith, J.A.; Hjorth, C.K.; Dommer, A.C.; Harbison, A.M.; Fogarty, C.A.; Barros, E.P.; Taylor, B.C.; McLellan, J.S.; et al. Beyond Shielding: The roles of glycans in the SARS-CoV-2 Spike Protein. ACS Cent Sci 2020, 6, 1722-1734, doi:10.1021/acscentsci.0c01056.
  4. Dankai, W.; Pongpom, M.; Vanittanakom, N. Validation of reference genes for real-time quantitative RT-PCR studies in Talaromyces marneffei. J Microbiol Methods 2015, 118, 42-50, doi:10.1016/j.mimet.2015.08.015.

Reviewer 2 Report

This manuscript presents a valuable overview of T. marneffei homeobox TFs and then drills down to focus on one, RfeB, that is implicated by expression data in the hyphal morphogenesis transition.  The study goes on to model the structure of RfeB and its possible interaction with prospective partner Swi5.  The logic is well grounded in precedents, notably from S. cerevisiae.

The manuscript is very well written, and manages to incorporate and explain structural detail witch clear logic.

I have two  criticisms.

1.  Less important.  It is a pity that SltA was not included in the study.  Even though it has been characterized previously, it would make this study a real cornerstone if expression data for all homeobox protein genes were included.  If the RNA samples have been preserved, this would not be much work.

2.  More important.  The QRT-PCR normalization control gene, encoding actin, is often used for fungi but is wholly inappropriate.  It has been known for decades that actin genes from, for example, S. cerevisiae and C. albicans are highly regulated.  You can see that here:

https://www.yeastgenome.org/locus/S000001855#regulation

and here:

https://fungidb.org/fungidb/app/record/gene/C1_13700W_A#TranscriptionSummary

So you should find another control.  Or even better, use a panel of controls.

Two minor things should be attended to.

Line 79-80.  It is said that "MATa1/alpha2 are the homeodomain proteins involved with mating type switching and sexual reproduction."  That is true, but their main role is in cell type determination rather than switching.  You can see that their role in switching is still not entirely understood:

https://journals.asm.org/doi/10.1128/microbiolspec.mdna3-0013-2014

I recommend replacing "mating type switching" with "cell type determination" in that sentence.

Line 152.  The Candida Genome Database should be cited so that its funders know that it is being used.  The relevant citation is:

Skrzypek MS, Binkley J, Binkley G, Miyasato SR, Simison M, Sherlock G. 2017. The Candida Genome Database (CGD): incorporation of Assembly 22, systematic identifiers and visualization of high throughput sequencing data. Nucleic Acids Res 45:D592-D596.

Author Response

Please also see the attachment file.

Comment 1: Less important.  It is a pity that SltA was not included in the study.  Even though it has been characterized previously, it would make this study a real cornerstone if expression data for all homeobox protein genes were included.  If the RNA samples have been preserved, this would not be much work.

Response 1: Our original aim was to test the expression of uncharacterized homeobox genes to gain insights into the biological relevance of bioinformatically identified genes. On the other hand, the TmStlA gene and its functions have been identified and well characterized by genetic study and expression analysis (Borneman et al. 2000). Moreover, the RNA samples from this study were ran out, and we cannot prepare new RNA samples within a 10-day revision timeframe, as it will take at least 10-14 days for T. marneffei to generate conidia. Therefore, to emphasize the critical functions of STE12 and its homologs, we generated Table 2 (revised manuscript) to summarize data from experimentally verified STE12 genes. The following modified sentence was added.

            (Page 2, line 70-72, revised manuscript) Strikingly, this homeodomain-containing transcription factor family plays a significant role in the regulation of sexual reproduction, morphogenesis, and metabolism across fungal members of Ascomycota and Basidiomycota (Table 1 and Table 2).

We are compared the TmStlA gene expression data by Borneman et al. 2000 with transcriptomic data reported by Yang et al. 2014. The following sentence was added into the main text.

            (Page 12, line 281, revised manuscript) For further analysis, we focused on seven uncharacterized genes that contain the homeobox domain IPR001356, designated as TmHox1 – TmHox7 (Table 4, Figure 3). We excluded the homeobox EEA26628.1 and EEA28535.1 proteins because they contain only the conserved site of the homeobox protein, antennapedia type (IPR001827) while the homeobox protein domain IPR001356 is absent (Figure 3, Supplemental Data S1). Also, the StlA gene was previously characterized in T. marneffei to be expressed primarily in vegetative tissues, and therefore excluded from our current study [37] (Table 5; Yang et al. 2014).

Comment 2: More important.  The QRT-PCR normalization control gene, encoding actin, is often used for fungi but is wholly inappropriate.  It has been known for decades that actin genes from, for example, S. cerevisiae and C. albicans are highly regulated.  You can see that here:

https://www.yeastgenome.org/locus/S000001855#regulation

and here:

https://fungidb.org/fungidb/app/record/gene/C1_13700W_A#TranscriptionSummary

So you should find another control.  Or even better, use a panel of controls.

Response 2: Since T. marneffei is a dimorphic fungus, different forms express their own basal level of genes. Therefore, this fungus is different than the monomorphic fungi and very difficult to find the proper internal control. However, our laboratory (Dankai et al. 2015 [4]) has previously determined four housekeeping genes that could be potentially used as endogenous control for expression analysis by qRT-PCR. These genes are as follows:  actin (act); glyceraldehyde-3-phosphate dehydrogenase (gapdh); β-tubulin (benA) and 18S rRNA. This study indicates that actin gene is the most suitable reference gene to use in T. marneffei qRT-PCR experiment across different cell types or stress conditions (Dankai et al. 2015 [4]). We added this reference and primer sequences for actin gene in Table 3.

Line 79-80.  It is said that "MATa1/alpha2 are the homeodomain proteins involved with mating type switching and sexual reproduction."  That is true, but their main role is in cell type determination rather than switching.  You can see that their role in switching is still not entirely understood:

https://journals.asm.org/doi/10.1128/microbiolspec.mdna3-0013-2014

I recommend replacing "mating type switching" with "cell type determination" in that sentence.

Response: We edited sentence as suggested.

Line 152.  The Candida Genome Database should be cited so that its funders know that it is being used.  The relevant citation is:

Skrzypek MS, Binkley J, Binkley G, Miyasato SR, Simison M, Sherlock G. 2017. The Candida Genome Database (CGD): incorporation of Assembly 22, systematic identifiers and visualization of high throughput sequencing data. Nucleic Acids Res 45:D592-D596.

Response: These references were added to the text and reference section.

Reference

  1. Kummasook, A.; Tzarphmaag, A.; Thirach, S.; Pongpom, M.; Cooper, C.R., Jr.; Vanittanakom, N. Penicillium marneffei actin expression during phase transition, oxidative stress, and macrophage infection. Mol Biol Rep 2011, 38, 2813-2819, doi:10.1007/s11033-010-0427-1.
  2. Borneman, A.R.; Hynes, M.J.; Andrianopoulos, A. An STE12 homolog from the asexual, dimorphic fungus Penicillium marneffei complements the defect in sexual development of an Aspergillus nidulans steA mutant. Genetics 2001, 157, 1003-1014, doi:10.1093/genetics/157.3.1003.
  3. Casalino, L.; Gaieb, Z.; Goldsmith, J.A.; Hjorth, C.K.; Dommer, A.C.; Harbison, A.M.; Fogarty, C.A.; Barros, E.P.; Taylor, B.C.; McLellan, J.S.; et al. Beyond Shielding: The roles of glycans in the SARS-CoV-2 Spike Protein. ACS Cent Sci 2020, 6, 1722-1734, doi:10.1021/acscentsci.0c01056.
  4. Dankai, W.; Pongpom, M.; Vanittanakom, N. Validation of reference genes for real-time quantitative RT-PCR studies in Talaromyces marneffei. J Microbiol Methods 2015, 118, 42-50, doi:10.1016/j.mimet.2015.08.015.